# Rationalizing and Augmenting Dynamic Graph Neural Networks

**Guibin Zhang**[1,2,*]**, Yiyan Qi**[2,*]**, Ziyang Cheng**[1]**, Yanwei Yue**[1]**, Dawei Cheng**[1✉]**, Jian Guo**[2✉]
[1]Tongji University    [2]International Digital Economy Academy (IDEA)
✉ Corresponding: `dcheng@tongji.edu.cn`, `guojian@idea.edu.cn`

## Abstract

Graph data augmentation (GDA) has shown significant promise in enhancing the performance, generalization, and robustness of graph neural networks (GNNs). However, contemporary methodologies are often limited to static graphs, whose applicability on dynamic graphs—more prevalent in real-world applications—remains unexamined. In this paper, we empirically highlight the challenges faced by static GDA methods when applied to dynamic graphs, particularly their inability to maintain temporal consistency. In light of this limitation, we propose a dedicated augmentation framework for dynamic graphs, termed `DyAug`, which adaptively augments the evolving graph structure with temporal consistency awareness. Specifically, we introduce the paradigm of graph rationalization for dynamic GNNs, progressively distinguishing between causal subgraphs (*rationale*) and the non-causal complement (*environment*) across snapshots. We develop three types of environment replacement, including, spatial, temporal, and spatial-temporal, to facilitate data augmentation in the latent representation space, thereby improving the performance, generalization, and robustness of dynamic GNNs. Extensive experiments on six benchmarks and three GNN backbones demonstrate that `DyAug` can **(I)** improve the performance of dynamic GNNs by $0.89\% \sim 3.13\%$ ↑; **(II)** effectively counter targeted and non-targeted adversarial attacks with $6.2\% \sim 12.2\%$ ↑ performance boost; **(III)** make stable predictions under temporal distribution shifts. The source code is available at `https://github.com/bingreeky/DyAug`.

## 1 Introduction

Data-driven inference has greatly enhanced generalization capabilities and improved model performance, particularly through data augmentation, across a wide range of domains, including computer vision (CV) (Yang et al., 2022; Alomar et al., 2023; Zheng et al., 2023), natural language processing (NLP) (Shorten et al., 2021; Dai et al., 2023), and graph-based tasks (Rong et al., 2019; Feng et al., 2020; Fang et al., 2022; Wang et al., 2021b; Liu et al., 2022c; Sui et al., 2024). Among these domains, graph-tailored augmentation strategies are notably distinct due to the irregular and non-Euclidean nature of graph data, unlike the other two domains, where data is typically structured in regular, Euclidean forms such as grids (*i.e.*, images) and sequences (*i.e.*, sentences). Drawing upon these unique characteristics, current graph data augmentation (GDA) techniques have demonstrated significant effectiveness in enhancing the performance (Zhao et al., 2022c; Wang et al., 2024; Li et al., 2023d), robustness (Jin et al., 2020; Kong et al., 2022; Zhang et al., 2024c;b), and generalization ability (Wu et al., 2022b; Liu et al., 2022a) on graph neural networks (GNNs), applicable to various levels of graph tasks, including node-level (Rong et al., 2019; Wang et al., 2021b; Zhang et al., 2024d; Li et al., 2025), edge-level (Dai et al., 2019; Zhao et al., 2022b), and graph-level (Feng et al., 2020; You et al., 2021; Liu et al., 2022a).

Despite their success, previous graph data augmentation methods have been largely constrained to a specific class of graphs, namely *static graphs*. In contrast, *dynamic graphs*, which evolve over time, and are widely recognized as more prevalent in real-world applications such as social networks (Berger-Wolf & Saia, 2006; Greene et al., 2010; Li et al., 2023c), financial transactions (Nascimento et al., 2021; Zhang et al., 2021), and traffic networks (Peng et al., 2021; 2020),

---

*Equal contributions.

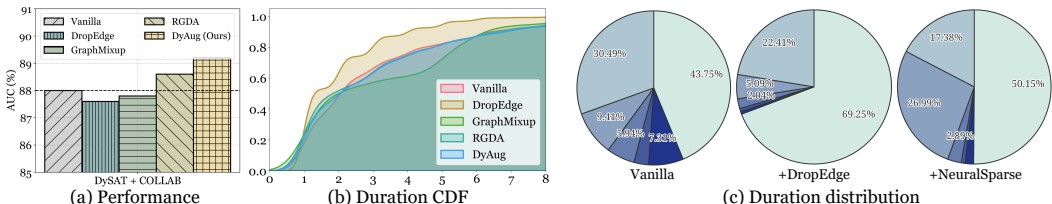

**Figure 1: (Left)** (a) illustrates the AUC (%) comparison among vanilla DySAT, DropEdge (Rong et al., 2019), GraphMixup (Wang et al., 2021b), RGDA (Liu et al., 2024a) and our `DyAug` on the `Yelp` dataset; **(Middle)** (b) analyzed the time span of each edge, *i.e.*, its longest consecutive existence period across snapshots, and compared the cumulative distribution function (CDF) for the vanilla dataset and after applying NeuralSparse and DropEdge; **(Right)** (c) visualizes the distribution of edge time spans under different methods, with colors ranging from light to dark representing $[1, 2, 3, 4, 5, \text{others}]$.

notably lack effective and tailored data augmentation designs. In light of this dilemma, an intuitive approach would be to directly apply static graph augmentation techniques to dynamic graphs. But *is this truly feasible?* In the remainder of this section, we will (1) first review key advancements in GDA, (2) highlight the critical challenges in their extension to dynamic graphs, (3) present our proposal for dynamic graph augmentation, and (4) summarize the contributions of this paper.

**Look-back on GDA.** Existing graph augmentation techniques can be broadly categorized into the following two types: ❶ **rule-based augmenters**, which manipulate data through predefined strategies such as edge dropping (Rong et al., 2019), node dropping (Feng et al., 2020), message dropping (Fang et al., 2022), and mixup (Wang et al., 2021b; Han et al., 2022; Ling et al., 2023; Kim et al., 2023; Ma et al., 2024); and ❷ **learning-based augmenters**, which incorporate learnable parameters in the generation of augmented examples, including graph structure learning (Jin et al., 2020; Wu et al., 2022a; Zou et al., 2023; Huang et al., 2023; Wan et al., 2024b), graph rationalization (Liu et al., 2022a; Wu et al., 2022b; Yue et al., 2024a), adversarial training (Zhu et al., 2019; Li et al., 2023a), and contrastive learning (You et al., 2020; 2021; Shen et al., 2023). Although these methods have made significant strides in *empowering*, *generalizing*, and *robustifying* GNNs on static graphs (Zhao et al., 2022a; Ding et al., 2022; Adjeisah et al., 2023), their performance on dynamic graphs , along with the potential challenges, remains largely unexplored and unassessed.

**Emperical Observation.** To investigate the potential issues of transferring static GDA methods to dynamic settings, we select three representative methods from both rule- and learning-based augmenters and apply them on `Yelp` (Sankar et al., 2020) with DySAT (Sankar et al., 2020) as the GNN backbone[1]. As shown in Figure 1 (*Left*), not all methods contribute positively: though effective on static graphs, DropEdge (Rong et al., 2019) and GraphMixup (Wang et al., 2021b), lead to a performance drop of $0.2\% \sim 0.6\%$. Nevertheless, RGDA (Liu et al., 2024a) achieves a modest improvement ($0.6\% \uparrow$ in AUC). This raises the question: *what caused such a discrepancy?* To explore this further, we examine how graph augmentation affects dynamic graph sequences. We investigate the edge timespan (Yang et al., 2023), namely the maximum number of consecutive graph snapshots an edge spans. As illustrated in Figure 1 (*Middle*), DropEdge significantly alters the cumulative distribution function (CDF) of edge timespans, while RGDA has a comparatively minor effect.

**Challenges.** Based on the above observations, we conclude that static GDA methods are not fully applicable to dynamic graphs, due to their unawareness of **temporal consistency**. More concretely, existing methods primarily focus on augmenting individual graphs, while overlooking the strong temporal dependencies across different graph snapshots. Imagine a scenario where existing GDA techniques are applied independently to each graph snapshot. Since topological augmentations often involve removing/deleting edges (Zhao et al., 2022a; Ding et al., 2022), this can lead to a surge in edges with a timespan of only 1, while many long-timespan edges may suddenly disappear at one timestamp and reappear at the next–such behavior does not align with the natural evolution patterns observed in real-world dynamic graphs (Beck et al., 2017). This issue is clearly illustrated in Figure 1 (*Right*): DropEdge disrupted many originally long-spanned edges, causing the proportion of edges with a timespan of 1 to increase sharply from $43.57\%$ to $69.25\%$. We call this the disruption of the *temporal consistency* (Qian et al., 2021) between different graph frames, which significantly undermines the continuity and temporal dependencies inherent in dynamic graphs. Therefore, a

---

[1] We place detailed explanations of their adaption to dynamic graphs in Appendix A.1.

natural question arises: *how can we design a data augmentation method for dynamic graphs that effectively enhances dynamic GNNs via well-maintaining temporal consistency?*

**Proposal.** To address the aforementioned challenges, we propose a *temporal-consistent dynamic graph augmentation framework* (dubbed `DyAug`), which can efficiently integrate into mainstream dynamic GNN backbones, and adaptively manipulates and augments evolving graph structures with temporal consistency awareness. Technically, `DyAug` pioneers the exploration of graph rationalization (Liu et al., 2022a; Si et al., 2023; Yue et al., 2024a) in dynamic graphs by dynamically pinpointing the rational features (*rationales*) that support a model's predictions during the training phase, while the contrasting elements are referred to as the *environment*. Through temporally conditioned graph rationale-environment separation, `DyAug` learns a highly correlated rationale subgraph sequence for the dynamic graphs, effectively preserving the temporal consistency. By employing three types of augmentation techniques, *i.e.*, temporal, spatial, and spatial-temporal environment replacement, `DyAug` efficiently expands the data distribution for dynamic GNNs, achieving a triple win in terms of *performance*, *robustness*, and *generalization capacity*.

Our contributions are summarized as follows:

❶ *Problem Identification.* We pioneer the exploration of graph data augmentation (GDA) within dynamic graphs. Through empirical evaluation, we demonstrate the poor transferability of traditional static GDA methods to dynamic graphs, identifying the root cause as the disruption of temporal consistency inherent to dynamic graph structures.

❷ *Pratical Solution.* We propose a temporal-consistent dynamic graph augmentation framework, termed `DyAug`, which makes *the first step* to explore graph rationalization within dynamic GNNs. By extracting rationale subgraphs in a temporally conditioned manner, `DyAug` effectively preserves the temporal consistency of dynamic graph sequences while efficiently augmenting training data through temporal, spatial, and spatial-temporal environment replacement.

❸ *Experimental Validation.* Extensive experiments on six benchmarks and four dynamic GNN backbones demonstrate that `DyAug` can **(I)** improve the performance of dynamic GNNs by $0.89\% \sim 3.13\% \uparrow$, surpassing state-of-the-art GDA methods by up to $2.8\%$; **(II)** effectively counter targeted and non-targeted adversarial attacks with $6.20\% \sim 12.22\% \uparrow$ performance boost; **(III)** make stable predictions under temporal distribution shifts.

## 2 RELATED WORK

**Dynamic Graph Neural Networks.** Dynamic Graphs find applications in a wide variety of disciplines, including social networks (Berger-Wolf & Saia, 2006; Greene et al., 2010), recommender systems (Li et al., 2020; Zhang et al., 2022a; Gong et al., 2024), epidemiology (Wan et al., 2024a; Tan et al., 2024; Liu et al., 2024c), *etc*. According to data types, current dynamic graphs can be primarily classified into *discrete-time dynamic graphs* (DTDG) and *continuous-time dynamic graphs* (CTDG) (Barros et al., 2021; Feng et al., 2024). This paper's main research scope focuses on DTDG, which consists of multiple discrete graph snapshots arranged in chronological order. Contemporary dynamic GNNs (DyGNNs) typically follow a framework where a spatial module processes different snapshots, and a temporal module aggregates information from various timestamps. Common categories include: **(1) Typical GNN-RNN DyGNNs**, which utilize a GNN module to handle individual snapshots and employ recurrent neural network (RNN) style modules to aggregate information across time, including STGCN (Yu et al., 2017), DySAT (Sankar et al., 2020), EvolveGCN (Pareja et al., 2020), TeMP (Wu et al., 2020), TFE-GNN (Zhang et al., 2023a), and SEIGN (Qin et al., 2023); **(2) Temporal-enhanced DyGNNs**, where specific design are adopted to better capture temporal dependencies, such as generative adversarial networks (GAN) in SGNN-GR (Wang et al., 2022) and spiking neural networks (SNN) in SpikeNet (Li et al., 2023b); **(3) Spatial-enhanced DyGNNs**, improving spatial modeling, such as GCRN (Seo et al., 2018) and TTGCN (Li et al., 2024a).

**Graph Data Augmentation** As discussed in Section 1, existing graph data augmentation (GDA) techniques can be broadly categorized into two types. **(1) Rule-based augmentors** employ heuristic rules for enhancing graph data across various dimensions, including topological-level (Rong et al., 2019; You et al., 2020; Wang et al., 2020; Sun et al., 2021; Yue et al., 2024b), feature-level (Zhao et al., 2021a; Sun et al., 2021), and label-level (Park et al., 2021; Han et al., 2022). **(2) Learning-based augmenters** employ various automated learning paradigms to expand graph data. For instance, graph structure learning (Zhu et al., 2021; Li et al., 2024b; Zhiyao et al., 2024) has been leveraged for enhancing graph topology by adding or removing edges (Jin et al., 2020; Liu et al.,

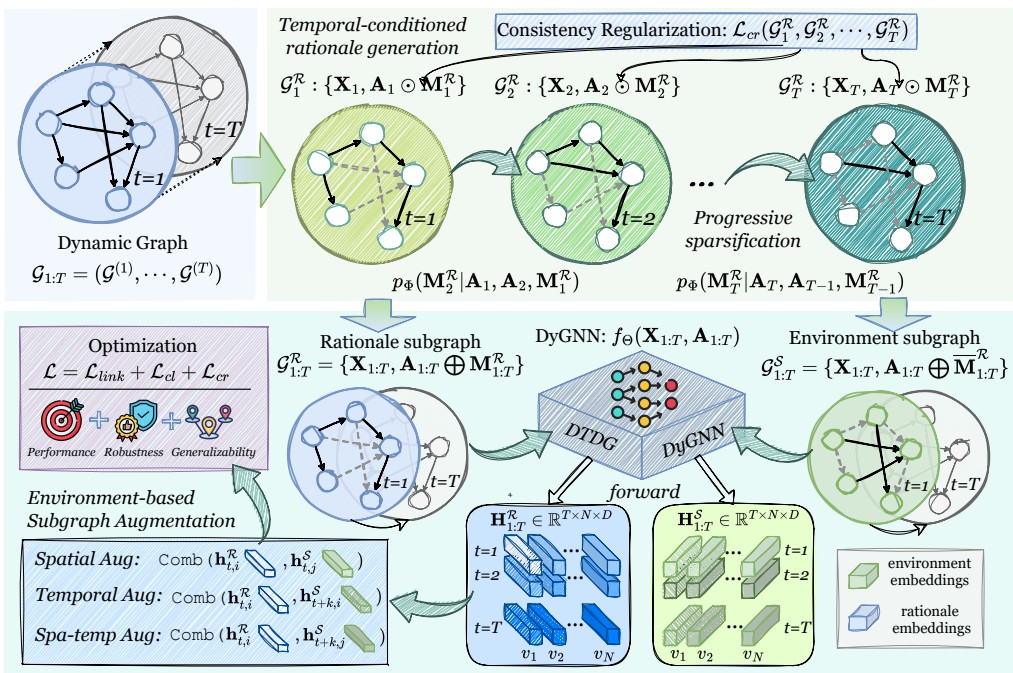

**Figure 2:** The overview of our proposed `DyAug`.

2022b; Zhao et al., 2023; Wu et al., 2023). Techniques such as contrastive learning (Velickovic et al., 2019; You et al., 2020; 2021; Zhang et al., 2023b; 2024e) and reinforcement learning (Zhao et al., 2022c; Zhou & Gong, 2023) are also commonly used for graph augmentation. In contrast to the aforementioned methods, graph rationalization focuses on identifying intrinsically learned subgraphs as the rationale graph (Zhao et al., 2022a), with the opposing context referred to as the environment, which augments data by perturbing the rationale-environment pair. Due to its robustness to data noise (Sun et al., 2022; Yuan et al., 2024) and distribution shifts (Wu et al., 2022b; Liu et al., 2024a), graph rationalization has garnered increasing attention, and our proposed method falls within this category. However, all these GDA methods are confined to static graphs and do not take into account the temporal correlations present in dynamic graph sequences, which significantly limits their performance when extended to dynamic graphs. Regarding GDA for dynamic graphs, although there are a few attempts for CTDG (Wang et al., 2021a; Chen et al., 2023), to the best of our knowledge, there currently are no GDA methods specifically designed for DTDG.

## 3 METHODOLOGY

### 3.1 NOTATIONS AND PRELIMINARY

**Notations** Consider a graph $\mathcal{G}$ with the node set $\mathcal{V}$ and the edge set $\mathcal{E}$. A (discrete-time) dynamic graph can be defined as $\mathcal{G}_{1:T} = (\mathcal{G}_1, \mathcal{G}_2, \cdots, \mathcal{G}_T)$, where $T$ is the number of time stamps, $\mathcal{G}_t = \{\mathcal{V}_t, \mathcal{E}_t\}$ is the graph slice at time stamp $t \in [1, T]$, $\mathcal{V} = \bigcup_{t=1}^{T} \mathcal{V}_t$, $\mathcal{E} = \bigcup_{t=1}^{T} \mathcal{E}_t$. Alternatively, $\mathcal{G}_{1:T}$ can be represented as $\{\mathbf{X}_{1:T}, \mathbf{A}_{1:T}\}$, where each snapshot $\mathcal{G}_t = \{\mathbf{X}_t, \mathbf{A}_t\}$ consists of node attributes $\mathbf{X}_t \in \mathbb{R}^{N \times D}$ and adjacency matrices $\mathbf{A}_t \in \{0, 1\}^{N \times N}$, with $N = |\mathcal{V}|$ denoting the number of nodes and $D$ denoting the dimensionality of the node attributes.

**DyGNN Paradigm** We take the classical *link prediction* in dynamic graph modeling as an example. The objective is to train a dynamic GNN $f_\Theta : \{\mathbf{X}_{1:T}, \mathbf{A}_{1:T}\} \longmapsto \{0, 1\}^{N \times N}$, which leverages information from the past $T$ snapshots to predict edge existence at time step $T + 1$. Furthermore, we can express $f_\Theta$ as $f_\Theta = f_d \circ f_t \circ f_s$, where $f_s : \{\mathbf{X}_{1:T}, \mathbf{A}_{1:T}\} \longmapsto \mathcal{H}_s$, parameterized by $\Theta_s$, is tasked with capturing spatial patterns to derive representations for timestamps $1 : T$, $f_t : \mathcal{H}_s \longmapsto \mathcal{H}_t$, parameterized by $\Theta_t$, captures temporal dependencies and retrieves representations for the $T + 1$ time step, and $f_d : \mathcal{H}_t \longmapsto \mathcal{Y}$, parameterized by $\Theta_d$, denotes the downstream task function, which predicts the connectivity $\mathbf{A}_{T+1}$ at time step $T + 1$ in the context of link prediction.

## 3.2 FRAMEWORK OVERVIEW

In this study, we present the first efficient data augmentation framework tailored for dynamic graphs, termed `DyAug`, as depicted in Figure 2. Given a dynamic graph sequence $\mathcal{G}_{1:T}$, `DyAug` progressively performs rationale-environment separation for each snapshot, constrained by consistency regularization to preserve temporal consistency across snapshots. The underlying DyGNN backbone is then employed to generate both rationale and environment representations. Subsequently, three types of augmentations, including spatial, temporal, and spatial-temporal, are applied to prevent $f_\Theta$ from learning spurious correlations from environment representations, thus enhancing the *performance*, *robustness*, and *generalizability* of DyGNNs. In the following sections, we will first give a causal analysis in Section 3.3, introduce how `DyAug` performs temporal-conditioned rationale-environment separation in Section 3.4, present the data augmentation strategies in Section 3.5, and showcase the overall optimization objective and complexity analysis in Section 3.6.

## 3.3 A CAUSAL VIEW ON DyGNNS

To clarify the goal and implementation of graph rationalization in dynamic graphs, we first take a step back to analyze the DyGNN modeling through a Structural Causal Model (SCM) (Pearl et al., 2000; Pearl, 2016), as illustrated in Figure 3. We depict the causal relationships among seven key variables in the DyGNN setting: the dynamic graph $\mathcal{G}_{1:T}$, the unobservable causal variable $\mathcal{C}$, the unobservable non-causal (environmental) variable $\mathcal{S}$, the observable node attributes $\mathbf{X}_{1:T}$, the observable topology $\mathbf{A}_{1:T}$, representations $\mathcal{H}$, and predictions $\mathcal{Y}$. Solid arrows indicate causal relationships, while dashed lines represent spurious correlations. Below are some critical insights about the SCM:

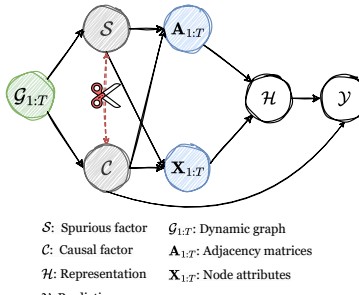

$\mathcal{S}$: Spurious factor  $\mathcal{G}_{1:T}$: Dynamic graph
$\mathcal{C}$: Causal factor  $\mathbf{A}_{1:T}$: Adjacency matrices
$\mathcal{H}$: Representation  $\mathbf{X}_{1:T}$: Node attributes
$\mathcal{Y}$: Prediction

**Figure 3:** Structural Causal Model (SCM) for DyGNNs.

- $\mathcal{S} \leftarrow \mathcal{G}_{1:T} \rightarrow \mathcal{C}$: The dynamic graph $\mathcal{G}_{1:T}$ consists of two disjoint parts: the causal part $\mathcal{C}$ and the non-causal/environmental part $\mathcal{S}$.

- $\mathcal{S} \rightarrow \mathbf{A}_{1:T} \leftarrow \mathcal{C}$ and $\mathcal{S} \rightarrow \mathbf{X}_{1:T} \leftarrow \mathcal{C}$: two variables (causal variable $\mathcal{C}$ and non-causal variable $\mathcal{S}$) construct two components of observable contextual subgraphs (node attributes $\mathbf{X}_{1:T}$ and topology $\mathbf{A}_{1:T}$), which is different from *i.i.d.* data only consider attributes.

- $\mathbf{A}_{1:T} \rightarrow \mathcal{H} \leftarrow \mathbf{X}_{1:T}$ and $\mathcal{H} \rightarrow \mathcal{Y}$: DyGNN backbones are leveraged to generate future representations based on observable contextual subgraphs, which are then utilized for making predictions.

- $\mathcal{C} \leftarrow - \rightarrow \mathcal{S}$: the spurious probabilistic dependencies between $\mathcal{S}$ and $\mathcal{C}$.

Based on this SCM, we can identify two backdoor paths between $\mathcal{C}$ and $\mathcal{Y}$: *(i)* $\mathcal{C} \leftarrow \mathcal{G}_{1:T} \rightarrow \mathcal{S} \rightarrow \mathbf{A}_{1:T} \rightarrow \mathcal{H} \rightarrow \mathcal{Y}$, and *(ii)* $\mathcal{C} \leftarrow \mathcal{G}_{1:T} \rightarrow \mathcal{S} \rightarrow \mathbf{X}_{1:T} \rightarrow \mathcal{H} \rightarrow \mathcal{Y}$. In both cases, the environmental variable $\mathcal{S}$ acts as a confounder between $\mathcal{G}_{1:T}$ and $\mathcal{Y}$, potentially causing a misleading correlation between $\mathcal{C}$ and $\mathcal{Y}$, if there is no direct causal link $\mathcal{C} \rightarrow \mathcal{Y}$. Therefore, severing $\mathcal{C} \leftarrow \rightarrow \mathcal{S}$ is crucial. In the following sections, we will elaborate on how `DyAug` effectively severs these spurious correlations through the rationale-environment separation and augmentation.

## 3.4 TEMPORAL-CONDITIONED RATIONALE–ENVIRONMENT SEPARATION

Given a graph snapshot $\mathcal{G}_t$, we model its unobservable causal part as a soft mask $\mathbf{M}_t^{\mathcal{R}} \in [0,1]^{N \times N}$, where each entry represents the probability score of the corresponding edge belonging to the rationale subgraph. Naturally, the rationale for snapshot $\mathcal{G}_t$ can be expressed as $\mathcal{G}_{(t)}^{\mathcal{R}} = \{\mathbf{X}_t, \mathbf{A}_t \odot \mathbf{M}_t^{\mathcal{R}}\}$, and the causal part of the entire dynamic graph sequence is denoted as $\mathcal{G}_{1:T}^{\mathcal{R}} = \{\mathbf{X}_{1:T}, \mathbf{A}_{1:T} \bigoplus \mathbf{M}_{1:T}^{\mathcal{R}}\}$. Traditional static graph rationalization methods (Zhang et al., 2024a; Liu et al., 2024a) typically follow the paradigm below to extract rationales:

$$\mathbf{M}_t^{\mathcal{R}} \sim p_\Phi(\mathbf{M}_t^{\mathcal{R}} \mid \mathbf{A}_t, \mathbf{X}_t), \tag{1}$$

where the causal mask is generated conditioned on the current features $\mathbf{X}_t$ and topology $\mathbf{A}_t$, and the generator is parameterized by $\Phi$. However, this overlooks the temporal dependencies between snapshots, often negatively disrupting the spatial-temporal distribution of the augmented data. To address this issue, we design **temporal-conditioned graph rationalization**, which progressively

uncovers the causal subgraph along the temporal dimension. At each step, it depends solely on the previous timestamp and current status, thereby maintaining the Markov property:

$$\mathbf{M}_t^{\mathcal{R}} \sim p_\Phi(\mathbf{M}_t^{\mathcal{R}} \mid \mathbf{A}_t, \mathbf{X}_t, \mathbf{M}_{t-1}^{\mathcal{R}}), \tag{2}$$

where the generation of $\mathbf{M}_t^{\mathcal{R}}$ additionally depends on the causal mask from the $(t-1)$-th timestamp. In practice, we model the rationale as follows:

$$p_\Phi(\mathbf{M}_t^{\mathcal{R}} \mid \mathbf{A}_t, \mathbf{X}_t, \mathbf{M}_{t-1}^{\mathcal{R}}) = \prod_{i=1}^{N} \prod_{j=1}^{N} p_\Phi(\mathbf{M}_{t,ij}^{\mathcal{R}} \mid \mathbf{A}_t, \mathbf{X}_t, \mathbf{M}_{t-1}^{\mathcal{R}}), \tag{3}$$

where $\mathbf{M}_{t,ij}^{\mathcal{R}}$ is the element at the $i$-th row and $j$-th column of $\mathbf{M}_t^{\mathcal{R}}$, representing the probability that edge $e_{ij}$ belongs to the rationale at timestamp $t$. We compute $\mathbf{M}_{t,ij}^{\mathcal{R}}$ as follows:

$$\begin{aligned} p_\Phi(\mathbf{M}_{t,ij}^{\mathcal{R}} = 1 \mid \mathbf{A}_t, \mathbf{X}_t, \mathbf{M}_{t-1}^{\mathcal{R}}) &= \mathbb{1}_{\mathbf{A}_t[i,j]=1} f_\Phi(\boldsymbol{x}_i^t, \boldsymbol{x}_j^t, \mathbf{M}_{t-1,ij}^{\mathcal{R}}), \\ &= \mathbb{1}_{\mathbf{A}_t[i,j]=1} \sigma\big((\log(\epsilon) - \log(1-\epsilon) + \varpi_{ij})/\tau\big), \end{aligned} \tag{4}$$

where $\mathbb{1}(\cdot)$ is an indicator function, $\omega_{ij} = \text{FFN}_\Phi([\boldsymbol{x}_i^t, \boldsymbol{x}_j^t, \mathbf{M}_{t,ij}^{\mathcal{R}}])$ parameterized by $\Phi$, $\epsilon \sim \text{Uniform}(0,1)$, $\sigma(\cdot)$ represents the sigmoid function, and $\tau$ is the temperature coefficient. When $\tau$ approaches zero, Equation (4) effectively returns the Bernoulli sampling result for edge $e_{ij}$. The gradient $\partial \mathbf{M}_{t,ij}^{\mathcal{R}}/\partial \varpi_{ij}$ remains well-defined as long as $\tau > 0$. Through this temporally progressive rationale discovery, we ultimately obtain the rationale subgraph set $\mathcal{G}_{1:T}^{\mathcal{R}}$ and the corresponding environmental subgraph set $\mathcal{G}_{1:T}^{\mathcal{S}}$, defined as follows:

$$\mathcal{G}_{1:T}^{\mathcal{R}} = \{\mathbf{X}_{1:T}, \mathbf{A}_{1:T} \oplus \mathbf{M}_{1:T}^{\mathcal{R}}\}, \ \mathcal{G}_{1:T}^{\mathcal{S}} = \{\mathbf{X}_{1:T}, \mathbf{A}_{1:T} \oplus \overline{\mathbf{M}}_{1:T}^{\mathcal{R}}\}, \tag{5}$$

where $\overline{\mathbf{M}}_t = \mathbf{A}_t - \mathbf{M}_t$. To further ensure temporal consistency, *i.e.*, the rationales across different snapshots remain coherent even as the graph structure evolves, we propose the following *consistency regularization loss*:

$$\mathcal{L}_{\text{cr}} = -\sum_{t=1}^{T} \sum_{p=t-w}^{t+w} \log \frac{\exp(\text{sim}(\mathcal{G}_t^{\mathcal{R}}, \mathcal{G}_p^{\mathcal{R}}))}{\exp\big(\text{sim}(\mathcal{G}_t^{\mathcal{R}}, \mathcal{G}_p^{\mathcal{R}})\big) + \sum_{k \notin [t-w,t+w]} \exp\big(\text{sim}(\mathcal{G}_t^{\mathcal{R}}, \mathcal{G}_k^{\mathcal{R}})\big)}, \tag{6}$$

where $\mathcal{G}_p^{\mathcal{R}}$ represents the rationale within a $w$-step temporal window surrounding the central rationale $\mathcal{G}_t^{\mathcal{R}}$, while $\mathcal{G}_k^{\mathcal{R}}$ refers to rationales that are further away. $\text{sim}(\cdot, \cdot)$ measures the similarity between graphs, and we implement it as $\text{sim}(\mathcal{G}_t^{\mathcal{R}}, \mathcal{G}_p^{\mathcal{R}}) = \text{sum}(|\mathbf{M}_t^{\mathcal{R}} - \mathbf{M}_p^{\mathcal{R}}|)$. Equation (6) draws inspiration from the practices in Tonekaboni et al. (2021); Wang et al. (2023), where closely situated subsequences are regarded as positive pairs and those with larger distance are treated as negatives. Exhibiting a similar idea, Equation (6) aims to maintain higher consistency around the current rationale. Upon disentangling the causal and non-causal components in dynamic graphs as well as maintaining the temporal consistency, we proceed to the next step of data augmentation.

### 3.5 DATA AUGMENTATION WITH ENVIRONMENT SUBGRAPHS

With the rationale subgraph set $\mathcal{G}_{1:T}^{\mathcal{R}}$ and the environmental subgraph set $\mathcal{G}_{1:T}^{\mathcal{S}}$ available, we explicitly separate the message passing from the rationale and environmental subgraphs during the spatial and temporal modeling processes of the vanilla DyGNN:

$$\mathbf{H}_{1:T}^{\mathcal{R}} = f_s(\mathbf{X}_{1:T}, \mathbf{A}_{1:T} \oplus \mathbf{M}_{1:T}^{\mathcal{R}}; \Theta_s), \ \mathbf{H}_{1:T}^{\mathcal{S}} = f_s(\mathbf{X}_{1:T}, \mathbf{A}_{1:T} \oplus \overline{\mathbf{M}}_{1:T}^{\mathcal{R}}; \Theta_s), \tag{7}$$

where $\mathbf{H}_{1:T}^{\mathcal{R}} \in \mathbb{R}^{T \times N \times D}$ and $\mathbf{H}_{1:T}^{\mathcal{S}} \in \mathbb{R}^{T \times N \times D}$ represent the aggregated rationale and environment embeddings for timestamps $1:T$. The standard $f_\Theta$ without rationalization tends to merge rationale and environment for predictions, leading to spurious correlations between $\mathcal{G}_{1:T}^{\mathcal{S}}$ and $\mathcal{Y}$, resulting in potentially accurate but unjustified predictions. Therefore, `DyAug` enhances the training samples by simulating various rationale-environment combinations within the node embedding space. To be more concrete, considering the rationale embedding $\mathbf{h}_{t,i}^{\mathcal{R}} = \mathbf{H}_t^{\mathcal{R}}[i,:]$ for node $v_i^t$ at timestamp $t$, we propose three environment replacement augmentation paradigms.

**Spatial Replacement Augmentation**    Data augmentation with environment replacement replaces the environment variable of the current node $v_i^t$ with any other environment variable from a different sample. Given that the rationable embedding $\mathbf{h}_{t,i}^{\mathcal{R}}$ is considered to be the dominant factor of predictions of $v_i$, the environment embedding can be interpreted as natural noise. Hence, combining $v_i^t$'s rationale embedding with environment embeddings from other nodes at the same snapshot enhances the model's robustness against the noise signal brought by the environment subgraphs. This process can be accomplished using any pooling function, such as concatenation, sum pooling, or max pooling. Taking sum pooling for example, the spatial replacement is executed as follows:

$$\hat{\mathbf{h}}_{t,i}^{\mathcal{R}} \leftarrow \texttt{Combine}(\mathbf{h}_{t,i}^{\mathcal{R}}, \mathbf{h}_{t,j}^{\mathcal{S}}) = \mathbf{h}_{t,i}^{\mathcal{R}} + \mathbf{h}_{t,j}^{\mathcal{S}}, \; j \sim \text{Uniform}(1, N) \setminus i, \tag{8}$$

where $\tilde{\mathbf{h}}_{t,i}^{\mathcal{R}}$ represents the augmented node embedding, and $\mathbf{h}_{t,j}^{\mathcal{S}} = \mathbf{H}_t^{\mathcal{S}}[j, :]$.

**Temporal Replacement Augmentation**    In contrast to spatial replacement, temporal enhancement focuses on augmenting the temporal dimension. As the dynamic graph structure evolves, $\mathbf{A}_{1:T} \oplus \mathbf{M}_{1:T}^{\mathcal{R}}$ remains a stable causal factor; however, $f_\Theta$ may inadvertently learn spurious patterns from the complementary non-causal components. Therefore, for each node $v_i$, we replace its environment embedding with one from a historical snapshot to avoid reliance on temporal trivial information:

$$\hat{\mathbf{h}}_{t,i} \leftarrow \texttt{Combine}(\mathbf{h}_{t,i}^{\mathcal{R}}, \mathbf{h}_{p,i}^{\mathcal{S}}), \; p \sim \text{Uniform}(1, t-1). \tag{9}$$

**Spatial-temporal Replacement Augmentation**    Combining the above two, spatial-temporal environment replacement augments data simultaneously across both spatial and temporal dimensions:

$$\hat{\mathbf{h}}_{t,i} \leftarrow \texttt{Combine}(\mathbf{h}_{t,i}^{\mathcal{R}}, \mathbf{h}_{p,j}^{\mathcal{S}}), \; (p, j) \sim \Big(\text{Uniform}(1, t-1), \text{Uniform}(1, N) \setminus i\Big). \tag{10}$$

After applying the three types of environment replacement, we obtain the augmented representations $\hat{\mathbf{H}}_{1:T} = [\hat{\mathbf{H}}_1, \hat{\mathbf{H}}_2, \ldots, \hat{\mathbf{H}}_T]$, where $\hat{\mathbf{H}}_t = [\hat{\mathbf{h}}_{t,1}, \hat{\mathbf{h}}_{t,2}, \ldots, \hat{\mathbf{h}}_{t,N}]$. These representations are then further transformed by the DyGNN backbone into future node embeddings $\hat{\mathbf{X}}_{T+1}$, which are subsequently converted into predictions $\hat{\mathcal{Y}}$ as follows:

$$\tilde{\mathbf{X}}_{T+1} = f_t(\tilde{\mathbf{H}}_{1:T}, \mathbf{A}_{1:T} \oplus \mathbf{M}_{1:T}^{\mathcal{R}}; \Theta_t), \; \hat{\mathcal{Y}} = f_d(\tilde{\mathbf{X}}_{T+1}; \Theta_d). \tag{11}$$

### 3.6    OPTIMIZATION AND ANALYSIS

**Optimization Objective**    In addition to the original task-specific loss associated with DyGNN training, denoted as $\mathcal{L}_{\text{pred}}$, DyAug introduces two supplementary losses: the *consistency regularization loss* defined in Equation (6), and the *contrastive loss* aimed at differentiating between causal and non-causal representations while ensuring semantic similarity between causal and augmented representations, which is expressed mathematically as follows:

$$\mathcal{L}_{\text{cl}} = \frac{1}{T} \sum_{t=1}^{T} \sum_{i=1}^{N} \log \frac{\exp(\text{sim}(\hat{\mathbf{h}}_{t,i}, \mathbf{h}_{t,i}^{\mathcal{R}})/\tau)}{\exp(\text{sim}(\hat{\mathbf{h}}_{t,i}, \mathbf{h}_{t,i}^{\mathcal{R}})/\tau) + \exp(\text{sim}(\mathbf{h}_{t,i}^{\mathcal{S}}, \mathbf{h}_{t,i}^{\mathcal{R}})/\tau)}, \tag{12}$$

where $\tau$ represents the temperature coefficient and $\text{sim}(\cdot, \cdot)$ is computed using the dot product. The overall training objective for DyAug is formulated as:

$$\min_{\Phi, \Theta} \mathcal{L}(\Phi, \Theta) = \mathcal{L}_{\text{pred}} + \alpha_1 \cdot \mathcal{L}_{\text{cr}} + \alpha_2 \cdot \mathcal{L}_{\text{cl}}, \tag{13}$$

where $\Phi$ pertains to rationale-environment separation, $\Theta$ parameterizes the vanilla DyGNN backbone, and $\alpha_1, \alpha_2$ are scaling factors.

**Complexity Analysis**    The additional complexity coming with DyAug arises primarily from three sources: (1) causal mask estimation, which contributes $\mathcal{O}(\sum_{t=1}^{T} |\mathcal{E}^{(t)}|D)$; (2) the contrastive loss, adding $\mathcal{O}(NDT^2)$; and (3) the consistency loss, contributing $\mathcal{O}(\varpi T \sum_{t=1}^{T} |\mathcal{E}^{(t)}|)$, which is negligible. In summary, the overall extra complexity introduced by DyAug is $\mathcal{O}(\sum_{t=1}^{T} |\mathcal{E}^{(t)}|D + NDT^2)$.

## 4    EXPERIMENTS

In this section, we conduct extensive experiments to answer the following research questions (**RQ**):
 **(RQ1)** Can DyAug augment dynamic GNNs?
 **(RQ2)** Can DyAug better preserve the temporal consistency of dynamic graphs?
 **(RQ3)** Does DyAug improve the robustness of DyGNNs against adversarial attacks?
 **(RQ4)** Does DyAug enhance the out-of-distribution generalization of DyGNNs?
 **(RQ5)** How sensitive is DyAug to its key components and parameters?

**Table 1:** AUC score (± standard deviation) of future link prediction task on five real-world datasets. The best results are in **bold**, and the runner-ups are underlined.

| | Method | COLLAB | Yelp | Bitcoin | UCI | ACT |
|---|---|---|---|---|---|---|
| GRCN | Vanilla | 0.8278±0.0052 | 0.6645±0.0187 | 0.8766±0.0034 | 0.7682±0.0074 | 0.7963±0.0041 |
| | +DropEdge | 0.8285±0.0059 | 0.6472±0.0205 | 0.8732±0.0051 | 0.7693±0.0105 | 0.7928±0.0042 |
| | +DropNode | 0.8297±0.0077 | 0.6448±0.0196 | 0.8749±0.0064 | 0.7664±0.0068 | 0.7942±0.0500 |
| | +DropMessage | 0.8326±0.0097 | 0.6669±0.0240 | 0.8780±0.0057 | 0.7690±0.0119 | 0.7968±0.0044 |
| | +GraphMixup | 0.8258±0.0049 | 0.6697±0.0180 | 0.8641±0.0038 | 0.7613±0.0133 | 0.7902±0.0045 |
| | +NeuralSparse | 0.8394±0.0076 | 0.6705±0.0233 | 0.8792±0.0065 | 0.7710±0.0062 | 0.7959±0.0046 |
| | +SUBLIME | 0.8312±0.0056 | 0.6685±0.0273 | 0.8770±0.0059 | 0.7735±0.0081 | 0.7989±0.0047 |
| | +RGDA | 0.8374±0.0031 | 0.6692±0.0194 | 0.8812±0.0031 | 0.7698±0.0046 | 0.8066±0.0048 |
| | +DyAug (ours) | **0.8495±0.0070** | **0.6795±0.0204** | **0.9079±0.0029** | **0.7783±0.0054** | **0.8147±0.0049** |
| DySAT | Vanilla | 0.8807±0.0018 | 0.7962±0.0045 | 0.8896±0.0027 | 0.7502±0.0056 | 0.7790±0.0036 |
| | +DropEdge | 0.8760±0.0039 | 0.7985±0.0106 | 0.8843±0.0048 | 0.7516±0.0077 | 0.7689±0.0067 |
| | +DropNode | 0.8783±0.0031 | 0.7980±0.0077 | 0.8826±0.0039 | 0.7519±0.0045 | 0.7705±0.0044 |
| | +DropMessage | 0.8815±0.0069 | 0.7964±0.0082 | 0.8915±0.0070 | 0.7574±0.0102 | 0.7689±0.0060 |
| | +GraphMixup | 0.8785±0.0112 | 0.7814±0.0029 | 0.8726±0.0031 | 0.7505±0.0030 | 0.7653±0.0012 |
| | +NeuralSparse | 0.8862±0.0053 | 0.8074±0.0050 | 0.8916±0.0044 | 0.7579±0.0064 | 0.7796±0.0061 |
| | +SUBLIME | 0.8871±0.0034 | 0.8037±0.0041 | 0.8927±0.0043 | 0.7552±0.0074 | 0.7772±0.0054 |
| | +RGDA | 0.8897±0.0030 | 0.8149±0.0059 | 0.8865±0.0036 | 0.7560±0.0051 | 0.7805±0.0041 |
| | +DyAug (ours) | **0.8925±0.0034** | **0.8233±0.0023** | **0.9032±0.0040** | **0.7698±0.0063** | **0.7860±0.0054** |
| SEIGN | Vanilla | 0.9219±0.0021 | 0.8072±0.0039 | 0.8955±0.0013 | 0.7932±0.0035 | 0.8457±0.0018 |
| | +DropEdge | 0.9235±0.0024 | 0.8015±0.0041 | 0.9015±0.0084 | 0.7985±0.0026 | 0.8366±0.0035 |
| | +DropNode | 0.9215±0.0039 | 0.8039±0.0046 | 0.8928±0.0035 | 0.7913±0.0015 | 0.8382±0.0030 |
| | +DropMessage | 0.9259±0.0032 | 0.8120±0.0032 | 0.8968±0.0071 | 0.7896±0.0024 | 0.8429±0.0018 |
| | +GraphMixup | 0.9187±0.0029 | 0.7942±0.0043 | 0.8905±0.0057 | 0.7903±0.0019 | 0.8381±0.0034 |
| | +NeuralSparse | 0.9279±0.0029 | 0.8038±0.0046 | 0.8994±0.0028 | 0.8012±0.0030 | 0.8491±0.0030 |
| | +SUBLIME | 0.9310±0.0027 | 0.8144±0.0015 | 0.8981±0.0025 | 0.8069±0.0020 | 0.8465±0.0013 |
| | +RGDA | 0.9245±0.0025 | 0.8131±0.0020 | 0.9013±0.0018 | 0.7985±0.0039 | 0.8472±0.0013 |
| | +DyAug (ours) | **0.9362±0.0022** | **0.8284±0.0015** | **0.9067±0.0017** | **0.8098±0.0022** | **0.8546±0.0029** |

## 4.1 EXPERIMENTAL SETUP

**Datasets and Splits** To thoroughly evaluate our proposed method, we select five real-world datasets. COLLAB (Tang et al., 2012) is an academic collaboration network spanning 16 years. Yelp (Sankar et al., 2020) is a business review dataset containing customer feedback on various businesses. Bitcoin (Kumar et al., 2018) is a trust network dataset representing users who engage in trading on the Bitcoin OTC platform. UCI (Panzarasa et al., 2009) is an online communication network from the University of California, Irvine, capturing student interactions. Lastly, ACT (Kumar et al., 2019) describes the actions taken by users on a popular MOOC website within 30 days.

**Backbones and Baselines** For DyGNN backbones, we select three classical baselines: **(1) GCRN** (Seo et al., 2018), which combines GCNs and GRU (Chung et al., 2014); **(2) DySAT** (Sankar et al., 2020), which models spatial and temporal dependencies via self-attention; and **(3) SEIGN** (Qin et al., 2023), which leverages GCN for message passing, GRU for parameter updates, and transformer for learning the final node representations. For baselines, we comprehensively choose seven GDA techniques for comparison. Regarding rule-based augmenters, we adopt **DropEdge** (Rong et al., 2019), **DropNode** (Feng et al., 2020), **DropMessage** (Fang et al., 2022), and **Graph Mixup** (Wang et al., 2021b). Regarding rule-based augmenters, we opt for **NeuralSparse** (Zheng et al., 2020) and **SUBLIME** (Liu et al., 2022d) for the *graph structure learning* branch, **RGDA** (Liu et al., 2024a) for the *graph rationalization* branch. Notably, we acknowledge that some classical or highly related GDA methods, such as DIR (Wu et al., 2022b), GREA (Liu et al., 2022a), JOAO (You et al., 2021), and AIA (Sui et al., 2024), are not included in our evaluation. This exclusion is either due to data format limitations (*e.g.*, methods focused solely on graph classification) or incompatibility issues (*e.g.*, inability to adapt to dynamic graphs). Detailed explanations and the full baseline setup can be found in Appendix A.1.

**Hyperparameter Configurations** We set the number of layers to two for all baselines, with a hidden dimension of 128. Specifically, for DyAug, we fix $\tau = 1e-2$ and $\varpi = 2$ across all datasets. For each node, we assign an equal probability of selecting one of the following strategies: (1) no replacement, using only the rationale embedding; (2) spatial replacement; (3) temporal replacement; or (4) spatial-temporal replacement. We provide an ablation study on the effectiveness of these replacement strategies in Appendix B.2. For the parameters $\alpha_1$ and $\alpha_2$ in Equation (13), we vary $\alpha_1 \in \{1e-2, 5e-2, 1e-1\}$ and $\alpha_2 \in \{1e-4, 5e-4, 1e-3, 5e-3\}$.

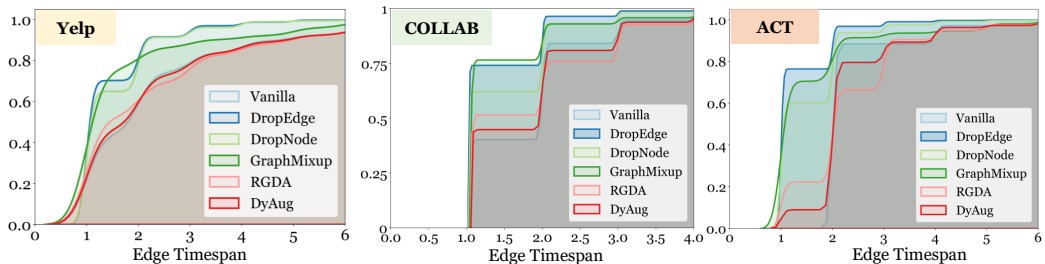

**Figure 4:** The cumulative distribution function (CDF) of edge timespan for the vanilla dataset and after applying different GDA methods on YELP, COLLAB and ACT. We opt for GRCN for all the datasets. The faster the curve converges to 1, the greater the proportion of edges with shorter timespans.

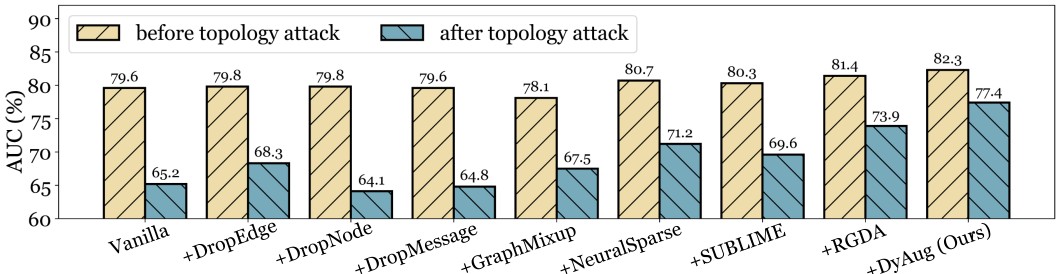

**Figure 5:** The performance comparison of various GDA methods under **structure attack** on DySAT+YELP.

## 4.2 COMPARISON WITH EXISTING GDA METHODS (RQ1 & RQ2)

We compare the performance of `DyAug` against seven GDA baselines across five datasets on GRCN, DySAT, and SEIGN, summarized in Table 1. We draw the following observations (**Obs.**):

**Obs.❶ Temporal consistency counts for dynamic graphs.** DropEdge, DropNode, DropMessage, and GraphMixup belong to rule-based augmenters, and many of them consistently result in performance drops rather than improvements. For example, GraphMixup shows a $1.25\%\downarrow$ decrease on GRCN+Bitcoin, and DropNode shows a $0.85\%\downarrow$ drop on DySAT+ACT. We attribute the underlying cause to the disruption of temporal consistency. As shown in Figure 4, both DropNode and GraphMixup lead to a rapid convergence of the edge timespan CDF to 1, which indicates that these methods cause many long-spanning edges to disappear, replacing them with numerous short-lived edges that quickly appear and vanish. In contrast, RGDA and our `DyAug` closely approximate the vanilla CDF, resulting in far more stable and significant performance improvements.

**Obs.❷ `DyAug` continuously enhances dynamic graph modeling.** `DyAug` demonstrates a stable performance improvement across all datasets and DyGNN backbones. Nevertheless, the magnitude of this improvement is tied to the complexity of the DyGNN backbone. For simpler backbones like GRCN, `DyAug` yields notable gains, such as a $2.17\%$ increase on COLLAB and a $3.13\%$ boost on Bitcoin. While the AUC improvements on more advanced backbones like SEIGN are relatively marginal, `DyAug` still achieves a $2.14\%\uparrow$ increase on YELP.

## 4.3 AGAINST ADVERSARIAL ATTACKS (RQ3)

In this section, we aim to empirically validate that `DyAug` can effectively defend against both non-targeted and targeted attacks to answer RQ3. We employ three types of attacks: (1) **structure attack**, which randomly perturbs $20\%$ of edges; (2) **feature attack**, where Gaussian noise is added to the node features; and (3) **Nettack (poisoning mode)**, where Nettack (Zügner et al., 2018) is applied to perturb the training set twice. The results are visualized in Figures 5, 7 and 8. We observe:

**Obs.❸ `DyAug` outperform all baselines under various attack modes.** Among the three types of attacks, Nettack, as a targeted attack, poses the greatest challenge to the robustness of DyGNNs. As shown in Figure 8, the vanilla DySAT suffers from a significant drop of $12.7\%$ in AUC, and even when combined with RGDA, it only recovers $2.1\%$ of the loss. However, `DyAug`, through its tailored data augmentation design, effectively mitigates the impact of noise and perturbations during training, resulting in an $8.2\%$ improvement in the attacked model's performance.

**Table 2:** AUC score (%) of different methods on real-world datasets. The best results are in bold and the second-best results are underlined. 'w/o DS' and 'w/ DS' denote test data with and without distribution shift.

| Model | COLLAB (ROC-AUC↑) | | YELP (ROC-AUC↑) | |
|---|---|---|---|---|
| Test Data | w/o DS | w/ DS | w/o DS | w/ DS |
| GRCN (Seo et al., 2018) | 82.78± 0.52 | 67.49 ± 0.73 | 66.45 ± 1.87 | 61.82 ± 3.39 |
| + DyAug | 84.95± 0.70 | 73.22 ± 0.56 | 67.95 ± 0.24 | 61.78 ± 2.76 |
| DySAT (Sankar et al., 2020) | 88.07± 0.18 | 75.59 ± 0.29 | 79.62 ± 0.65 | 65.80 ± 1.22 |
| + DyAug | 89.25± 0.74 | 82.56 ± 0.68 | 82.33± 1.13 | 73.15 ± 0.92 |
| SEIGN (Qin et al., 2023) | 92.19± 0.21 | 80.68 ± 0.72 | 80.72 ± 0.39 | 67.19 ± 0.84 |
| + DyAug | **93.62± 0.42** | **83.11 ± 0.56** | **82.84 ± 0.35** | **76.50 ± 0.65** |
| IRM (Rosenfeld et al., 2021) | 87.96± 0.90 | 75.42 ± 0.87 | 66.49 ± 0.78 | 56.02 ± 6.08 |
| DIDA (Zhang et al., 2022b) | 91.97± 0.05 | 81.87 ± 0.40 | 78.22 ± 0.40 | 75.92 ± 0.90 |
| DGIB-Bern (Yuan et al., 2024) | 92.17± 0.20 | 83.09 ± 0.56 | 76.88 ± 0.20 | 72.56 ± 0.74 |

## 4.4 AGAINST DISTRIBUTION SHIFT (RQ4)

Graph rationalization was originally introduced to address the out-of-distribution (OOD) challenge by capturing invariant patterns in evolving data (Wu et al., 2021; 2024). To evaluate whether `DyAug` can defend against distribution shifts in dynamic graphs, we use COLLAB and Yelp, and explicitly construct distribution shifts. Specifically, for COLLAB, we transfer all edges belonging to "data mining" category to the test set, ensuring that DyGNN has never been exposed to this category during training. For Yelp, we select "Pizza" edges as the out-of-distribution data. We observe that:

**Obs.❹ `DyAug` effectively enhances the robustness of DyGNNs.** As demonstrated in Table 2, SEIGN+`DyAug` consistently achieves the best performance on both Yelp and COLLAB, regardless of the presence of distribution shifts. This improvement can be attributed not only to SEIGN's strong baseline performance but also to `DyAug`'s ability to sever spurious correlations. For example, on YELP, `DyAug` boosts SEIGN's OOD performance from 67.19% to 76.50%.

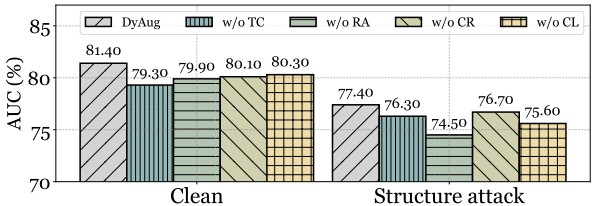

| $\alpha_2/\alpha_1$ | $1e-2$ | $5e-2$ | $1e-1$ |
|---|---|---|---|
| $1e-4$ | $78.14_{\pm0.2}$ | $\mathbf{78.60_{\pm0.2}}$ | $78.30_{\pm0.3}$ |
| $5e-4$ | $78.47_{\pm0.1}$ | $78.25_{\pm0.5}$ | $78.58_{\pm0.3}$ |
| $1e-3$ | $77.60_{\pm0.4}$ | $78.45_{\pm0.3}$ | $77.92_{\pm0.5}$ |
| $5e-3$ | $77.52_{\pm0.4}$ | $77.81_{\pm0.3}$ | $77.06_{\pm0.4}$ |

**Figure 6:** Ablation study on `DyAug` and its four variants, tested on ACT and that after structure attack. We use GRCN as the backbone.

**Table 3:** Sensitivity study on scaling factor $\alpha_1$ and $\alpha_2$. The results are reported on DySAT backbone and ACT dataset.

## 4.5 ABLATION STUDY AND SENSITIVITY ANALYSIS (RQ5)

**Ablation Study** We evaluate four variants: **(1)** `DyAug` ***w/o* TC**, where graph rationalization is snapshot-independent rather than temporally conditioned; **(2)** `DyAug` ***w/o* RA**, where all data augmentations are discarded; **(3)** `DyAug` ***w/o* CR**, where $\mathcal{L}_{cr}$ is omitted; and **(4)** `DyAug` ***w/o* CL**, where $\mathcal{L}_{cl}$ is discarded. It can be observed from Figure 6 that the removal of each module results in AUC decay, among which `DyAug` *w/o* RA is the most detrimental, leading to 2.9% ↓ drop under the structure attack. This demonstrates each component's importance in dynamic graph augmentation.

**Sensitivity Analysis** We evaluate `DyAug` under different $\alpha_1$ and $\alpha_2$. As shown in Table 3, `DyAug` is relatively insensitive to changes in $\alpha_1$. However, when $\alpha_2$ becomes too large, a performance drop is observed. For instance, with $\alpha_1 = 1e-2$, the performance with $\alpha_2 = 1e-4$ is 0.75% higher than that with $\alpha_2 = 5e-3$. Nevertheless, `DyAug` demonstrates overall robustness to parameter variations.

## 5 CONCLUSION

In this work, we present the first graph data augmentation (GDA) method specifically designed for (discrete-time) dynamic graphs, termed `DyAug`. `DyAug` addresses the limitations of previous static GDA methods, *i.e.*, the unawareness of temporal consistency, by employing temporal-conditioned rationale discovery to disentangle the rationale-environment within dynamic graph sequences. We further propose three augmentation strategies to enrich the data distribution. Experimental results demonstrate that `DyAug` excels in *empowering*, *robustifying*, and *generalizing* dynamic GNNs.

ACKNOWLEDGEMENT

Dawei Cheng is supported by the National Natural Science Foundation of China (62472317). Yiyan Qi is supported in part by the National Natural Science Foundation of China (62372362).

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

## A EXPERIMENTAL SETUP

### A.1 BASELINE SETUPS

Since many of the baselines we selected are originally designed for static graphs, in this section, we outline how these methods are adapted for dynamic graphs.

**Stochastic methods** DropEdge, DropNode, and DropMessage belong to stochastic dropping methods, which can be naturally integrated into the spatial modeling module of DyGNNs, as all three backbones use GNNs to capture spatial correlations.

**Mixup methods** Graph Mixup is a two-stage approach. After fully training the original DyGNN, we freeze its temporal module and continue training the spatial module following the Mixup paradigm from Wang et al. (2021b).

**Graph Structure Learning** Both NeuralSparse and SUBLIME are graph structure learning methods for GDA. For NeuralSparse, we initialize a denoising network for each graph snapshot and co-train it with DyGNN. For SUBLIME, we ensure that both $\mathbf{X}_{1:T}$ and $\mathbf{A}_{1:T}$ are visible to the model.

**Graph Rationalization** RGDA operates at both the graph and node levels. We apply its node-level augmentation independently for each snapshot and use sum pooling for $\texttt{Combine}(\cdot, \cdot)$ in RGDA.

### A.2 BASELINES NOT CHOOSED

We acknowledge that some classical or highly relevant GDA methods are not included in our evaluation. This is primarily due to two main reasons:

- **Task specificity to graph-level tasks**: Many graph rationalization methods are designed exclusively for graph-level tasks, including DIR (Wu et al., 2021), GREA (Liu et al., 2022a), GraphAug (Luo et al., 2023), DCT (Liu et al., 2024b), AIA (Sui et al., 2024), and C2R (Yue et al., 2024a). Thus, they were not selected for our evaluation.
- **Limited to static graphs**: When adapting static GDA methods to dynamic graphs, we faced considerable difficulties, especially with approaches that are inherently designed for individual graphs. For instance, GAUG-O (Zhao et al., 2021b) perturbs the original graph to generate $K$ perturbed graphs, which causes issues for DyGNN's temporal module when aggregating across different snapshots, as it cannot determine which perturbed snapshot to select. Consequently, we had to exclude several classical yet incompatible GDA baselines, such as Graph Transparent (Park et al., 2021), AutoGDA (Zhao et al., 2022c) and FLAG (Liu et al., 2022c).

## B SUPPLEMENTARY EXPERIMENT RESULTS

### B.1 RESULTS FOR RQ3

In Figure 7, we present the performance comparison of various GDA methods under a **feature attack**, where Gaussian noise is added to node features, Similarly, Figure 8 shows the performance comparison under the **Nettack**.

### B.2 RESULTS FOR ABLATION STUDY

To further compare the impact of different environment augmentation strategies within `DyAug`, we designed three variants: using only spatial replacement, using only temporal replacement, and using only spatial-temporal replacement. The results, as shown in Table 4, demonstrate that the variant with only spatial-temporal replacement closely approximates the performance of the full `DyAug`. On the `Yelp` and `ACT` datasets, `DyAug` (spa.) exhibits a significant performance drop, which is likely due to the longer edge timespans in these datasets (as shown in Figure 4), indicating a higher need for temporal augmentation.

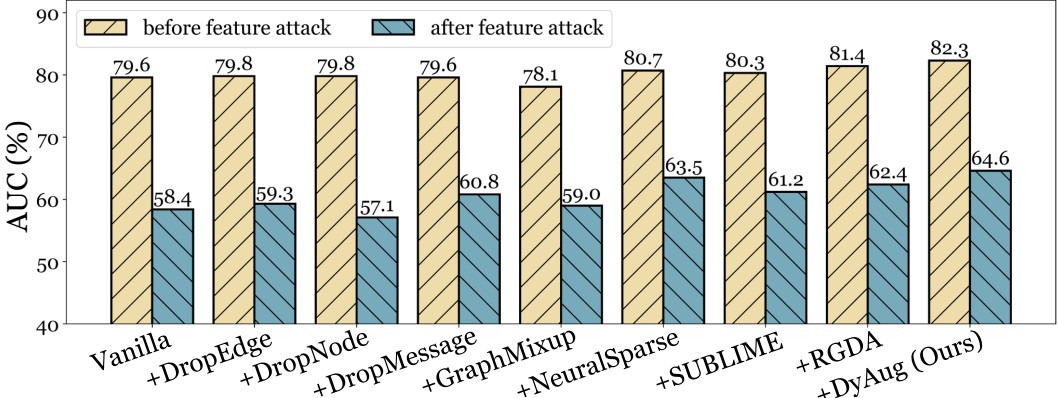

**Figure 7:** The performance comparison of various GDA methods under **feature attack** on DySAT+YELP.

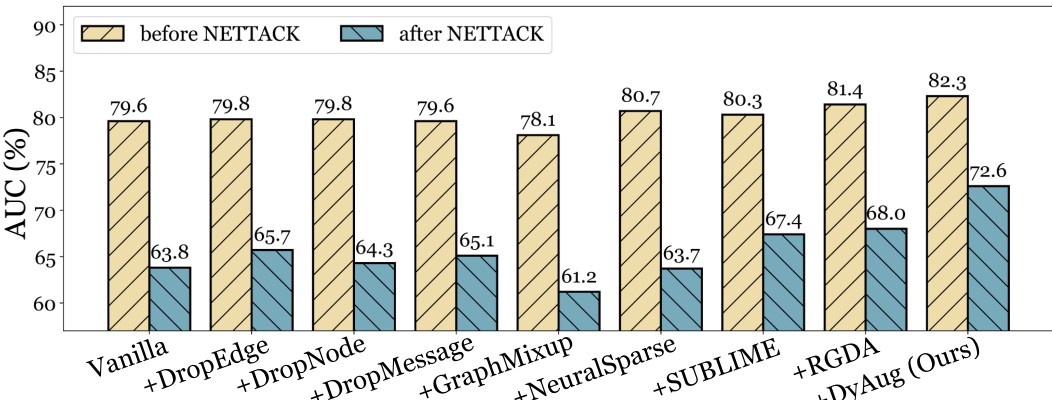

**Figure 8:** The performance comparison of various GDA methods under **Nettack** on DySAT+YELP.

**Table 4:** AUC score (%) of `DyAug` and its variants equipped with different augmentation strategies on real-world datasets. We select DySAT as the backbone. (tem.), (spa.), and (spa-tem.) denote the configurations where only temporal replacement, only spatial replacement, and only spatial-temporal replacement are applied.

| Model | COLLAB | Yelp | Bitcoin | ACT |
|---|---|---|---|---|
| DyAug | **0.8925±0.0074** | **0.8233±0.0043** | **0.9032±0.0060** | **0.7860±0.0084** |
| DyAug (tem.) | 0.8923±0.0047 | 0.8217±0.0047 | 0.8960±0.0052 | 0.7869±0.0060 |
| DyAug (spa.) | 0.8841±0.0050 | 0.8193±0.0065 | 0.8938±0.0052 | 0.7725±0.0043 |
| DyAug (tem-spa.) | 0.8937±0.0068 | 0.8215±0.0040 | 0.9079±0.0068 | 0.7810±0.0072 |

In addition, Tables 5 and 6 provides supplementary ablation studies under the scenarios of distribution shift, feature attack, and Nettack.

### B.3 RESULTS FOR SENSITIVITY ANALYSIS

To provide a more comprehensive evaluation of `DyAug`, we have included a sensitivity analysis on the window size $\varpi$ using the COLLAB and Yelp datasets, as shown in Table 7:

It can be observed that DyAug performs consistently well when $\varpi \in \{2, 4\}$, while for $\varpi = 8$, there is a notable performance drop on the COLLAB dataset (1.50% ↓ compared to $\varpi = 2$). We attribute this to the overly large window size, as the COLLAB dataset consists of only 16 snapshots. With such a large window, the majority of the dynamic graph sequence is considered as "positive pairs," rendering $\mathcal{L}_{cr}$ ineffective. Therefore, we have fixed $\varpi = 2$ for all subsequent experiments.

**Table 5:** Ablation study under **distribution shift**. The backbone is set as GRCN, and the dataset is COLLAB.

| Variant | DyAug | DyAug w/o TC | DyAug w/o RA | DyAug w/o CR | DyAug w/o CL |
|---------|-------|--------------|--------------|--------------|--------------|
| ROC-AUC | 73.22 | 72.76 | 70.58 | 73.08 | 72.70 |

**Table 6:** Ablation study under **topological attack** and **Nettack**. The backbone is set as GRCN, and the dataset is ACT.

| Variant | DyAug | DyAug w/o TC | DyAug w/o RA | DyAug w/o CR | DyAug w/o CL |
|---------|-------|--------------|--------------|--------------|--------------|
| Feature attack | 72.35 | 69.58 | 65.18 | 70.95 | 70.51 |
| Nettack | 74.68 | 72.05 | 71.41 | 74.36 | 73.05 |

**Table 7:** Sensitivity analysis on parameter $\varpi$ with GRCN backbone.

| Dataset | COLLAB | Yelp |
|---------|--------|------|
| $\varpi = 2$ | $0.8495 \pm 0.0070$ | $0.6795 \pm 0.0204$ |
| $\varpi = 4$ | $0.8506 \pm 0.0082$ | $0.6785 \pm 0.0139$ |
| $\varpi = 6$ | $0.8469 \pm 0.0069$ | $0.6732 \pm 0.0188$ |
| $\varpi = 8$ | $0.8345 \pm 0.0112$ | $0.6690 \pm 0.0228$ |

### B.4 RESULTS ON LARGE-SCALE GRAPHS

We extended our experiments to include a large-scale dynamic graph dataset, DGraphFin (Huang et al., 2022), using the slicing and data splitting strategy outlined in (Feng et al., 2024). This dataset comprises 4,889,537 nodes and 4,300,999 edges, as detailed in Table 8. We believe this benchmark effectively tests DyAug's scalability.

**Table 8:** Dataset characteristics.

| Dataset | COLLAB | Yelp | Bitcoin | UCI | ACT | DGraphFin |
|---------|--------|------|---------|-----|-----|-----------|
| #Nodes | 23,035 | 13,095 | 5,881 | 1,899 | 20,408 | 4,889,537 |
| #Edges | 151,790 | 65,375 | 35,591 | 59,835 | 202,339 | 4,300,999 |
| #Time steps | 16 | 24 | 21 | 13 | 30 | 12 |

We evaluated DyAug's performance on DGraphFin using DySAT as the backbone model, as shown in Table 9. From Table 9, we can conclude that DyAug seamlessly scales to ultra-large graphs. Graph structure learning methods like NeuralSparse and SUBLIME impose substantial GPU memory demands with marginal or even negative performance gains (2.46% ↓ for NeuralSparse). Rule-based augmenters such as DropEdge and DropNode introduce negligible computational overhead but fail to deliver notable performance gains, consistent with the findings in Table 1. DyAug, however, achieves a 1.72% AUC improvement with an additional GPU memory burden of less than 2 GB, demonstrating its exceptional scalability.

### B.5 RESULTS ON TD-PGD ATTACKS

We conducted additional experiments to evaluate DyAug under the more advanced attack method TD-PGD Sharma et al. (2023), as shown in Table 10. The results demonstrate that DyAug maintains strong robustness even under TD-PGD attacks. Notably, at $\epsilon = 0.1$, DyAug even achieves a performance improvement of 0.14% when applied on DySAT+UCI. What's more, DyAug demonstrated significant robustness on the DySAT+ACT setting under the TD-PGD attack, as it successfully improved the attacked performance from 68.53% to 74.90% at $\epsilon = 0.3$, achieving a remarkable 6.37% gain, which we believe serves as compelling evidence of its resilience in more challenging scenarios.

**Table 9:** Results on DGraphFin+DySAT. The implementation of DySAT is from Amazon-TGL (https://github.com/amazon-science/tgl). The results are reported on a single NVIDIA Tesla A100 40G GPU.

| Metric | AUC (%) | Per-epoch time (s) | GPU Memory (GB) |
|---|---|---|---|
| Vanilla | 73.26 | 47s | 3.7GB |
| +DropEdge | 72.69 | 43s | 3.7GB |
| +DropNode | 73.50 | 41s | 3.7GB |
| +DropMessage | 71.22 | 53s | 3.7GB |
| +NeuralSparse | 70.80 | 186s | 17GB |
| +SUBLIME | 73.15 | 147s | 21GB |
| +RGDA | 74.12 | 69s | 6.9GB |
| +DyAug | **74.98** | 56s | 5.6GB |

**Table 10:** Results tested with TD-PGD attack.

| Data | Method | $\epsilon = 0.0$ | $\epsilon = 0.1$ | $\epsilon = 0.2$ | $\epsilon = 0.3$ |
|---|---|---|---|---|---|
| UCI | DySAT | 0.7502±0.0056 | 0.7418±0.0092 | 0.7276±0.0143 | 0.7191±0.0129 |
| | +DyAug | 0.7698±0.0063 | 0.7712±0.0080 | 0.7456±0.0120 | 0.7466±0.0174 |
| ACT | DySAT | 0.7790±0.0036 | 0.7698±0.0028 | 0.7334±0.0138 | 0.6853±0.0262 |
| | +DyAug | 0.7860±0.0054 | 0.7834±0.0112 | 0.7682±0.0109 | 0.7490±0.0176 |

## B.6 RESULTS WITH MORE ADVANCED DYGNN BACKBONES

To further validate the wide applicability of DyAug, we have incorporated two advanced DyGNN backbones, Roland You et al. (2022) and SLATE Karmim et al. (2024). The experimental results are presented in Table 11. We observe that DyAug maintains excellent generalizability across these backbones, achieving a 1.0% and 1.87% performance gain on Roland and SLATE, respectively.

**Table 11:** Results on the UCI dataset. Roland is implemented with Roland-GRU.

| Backbone | Roland | | SLATE | |
|---|---|---|---|---|
| | ROC-AUC (%) | Gain | ROC-AUC (%) | Gain |
| Vanilla | 79.53±0.6122 | - | 81.26±0.3051 | - |
| +DropEdge | 79.25±0.4822 | $-0.28 \downarrow$ | 80.12±0.4715 | $-1.14 \downarrow$ |
| +DropNode | 79.29±0.6590 | $-0.24 \downarrow$ | 80.59±0.3552 | $-0.67 \downarrow$ |
| +DropMessage | 79.72±0.3145 | $+0.19 \uparrow$ | 81.77±0.5819 | $+0.51 \uparrow$ |
| +GraphMixup | 77.50±0.8911 | $-2.03 \downarrow$ | 80.35±0.7401 | $-0.91 \downarrow$ |
| +NeuralSparse | 79.86±0.4818 | $+0.33 \uparrow$ | 81.75±0.4291 | $+0.49 \uparrow$ |
| +SUBLIME | 80.23±0.5741 | $+0.70 \uparrow$ | 82.56±0.3069 | $+1.30 \uparrow$ |
| +RGDA | 79.86±0.9582 | $+0.33 \uparrow$ | 82.77±0.3593 | $+1.51 \uparrow$ |
| +DyAug | **80.53±0.7318** | $+1.00 \uparrow$ | **83.13±0.2639** | $+1.87 \uparrow$ |

