# OpenReview forum: "Rationalizing and Augmenting Dynamic Graph Neural Networks"
_ICLR.cc/2025/Conference — ICLR 2025 Poster_

### Official Review · Reviewer_nnH6 · 2024-10-23

**Soundness:** 3
**Presentation:** 3
**Contribution:** 2
**Rating:** 6
**Confidence:** 4

**Summary:**

This paper proposes a novel data augmentation method designed for dynamic graphs. Previous static graph data augmentation methods have shown poor performance on dynamic graphs, a phenomenon the authors attribute to the direct application of static graph augmentation techniques, which neglects the temporal consistency of graph structures. The authors introduce a learnable data augmentation method for dynamic graphs, incorporating a consistency loss function to ensure the temporal consistency of graph structures. Additionally, to enhance the performance of this data augmentation method in out-of-distribution generalization scenarios, the authors adopt a causal inference perspective to partition the original graph into rationale and environment subgraphs, thereby improving the algorithm's stability under data distribution shifts.

**Strengths:**

1. The authors present a new data augmentation approach motivated by the intuitive notion of temporal consistency in graph structures, which consistently achieves favorable performance in dynamic graph tasks. Furthermore, this method demonstrates stable augmentation in scenarios such as adversarial attacks and out-of-distribution generalization, showcasing the robustness of the algorithm.
2. The authors conducted extensive experiments based on various baselines, illustrating that this data augmentation method consistently provides enhancement across different baselines and scenarios.

**Weaknesses:**

1. The authors omitted ablation experiments in out-of-distribution generalization scenarios and adversarial attack scenarios other than structure attacks, which reduces the persuasiveness of the algorithm's effectiveness explanation.
2. The complexity analysis presented in section 3.6. Specifically, during the computation of the consistency loss, the complexity associated with determining the similarity between two graphs is not constant. Instead, it is plausible that this complexity scales linearly with respect to the number of edges. Consequently, the overall complexity might be revised to $\mathcal{O}(wT|\mathcal{E}|)$.
3. More recent works about dynamic graphs should be properly included in related works or compared as baselines.

**Questions:**

1. The source code repo is expired.
2. In section 3.4, the description of $\overline{M}=1_N-M$ and the specification that $1_N\in \set{1}^{N\times N}$ is an all-one matrix is somewhat perplexing. Given the typical sparsity of graphs, the transformation $\overline{M}=1_N-M$ would result in a very dense matrix, thereby significantly increasing computation as the number of edges grows. This raises the question of whether the intended expression was $\overline{M}=A-M$, where $A$ represents the adjacency matrix of the graph, rather than an all-one matrix.

---

> ### Author Response · Authors · 2024-11-20
> **[Part 1/2] Response to Reviewer nnH6**
>
> We are truly grateful for the time you have taken to review our paper and your insightful review. Here we address your comments in the following.
>
> ---
> > `Weakness 1` The authors omitted ablation experiments in out-of-distribution generalization scenarios and adversarial attack scenarios other than structure attacks, which reduces the persuasiveness of the algorithm's effectiveness explanation.
>
>
> Thank you for your insightful feedback, which has significantly strengthened our work! To address your concerns, we conducted three sets of ablation studies under different scenarios: (1) **distribution shift**, (2) **feature attacks**, and (3) **Nettack attacks**. The results are summarized below:
>
> _Table A. Ablation study under **distribution shift**. The backbone is set as GRCN, and the dataset is COLLAB._
> |Variant|DyAug|DyAug w/o TC| DyAug w/o RA| DyAug w/o CR|  DyAug w/o CL|
> |-|-|-|-|-|-|
> |ROC-AUC|73.22|72.76|70.58|73.08|72.70|
>
> _Table B. Ablation study under **topological attack** and **Nettack**. The backbone is set as GRCN, and the dataset is ACT._
> |Variant|DyAug|DyAug w/o TC| DyAug w/o RA| DyAug w/o CR|  DyAug w/o CL|
> |-|-|-|-|-|-|
> |feature attack|72.35|69.58|65.18|70.95|70.51|
> |Nettack|74.68|72.05|71.41|74.36|73.05
>
> From Tables A and B, we observe that removing **temporal-conditioned rationalization (TC)** and **data augmentation (RA)** consistently leads to the most significant performance drop across all scenarios. The impact of **$\mathcal{L}_{cr}$** (consistency regularization loss) and **$\mathcal{L}_{cl}$** (contrastive learning loss) varies by scenario. For instance, in the case of Nettack or structural attacks, removing $\mathcal{L}_{cl}$ results in a larger degradation than removing $\mathcal{L}_{cr}$.
>
> We have updated these ablation results in the manuscript and hope they adequately address your concerns!
>
>
> ---
> > `Weakness 2` The complexity analysis presented in section 3.6. Specifically, during the computation of the consistency loss, the complexity associated with determining the similarity between two graphs is not constant. Instead, it is plausible that this complexity scales linearly with respect to the number of edges. Consequently, the overall complexity might be revised to $\mathcal{O}(\varpi T|\mathcal{E}|)$
>
> Thank you once again for your meticulous review! Following your suggestion, we have updated the complexity analysis in Section 3.6 of the revised manuscript to reflect the corrected complexity of $\mathcal{O}(\varpi T|\mathcal{E}|)$.

---

> ### Author Response · Authors · 2024-11-20
> **[Part 2/2] Response to Reviewer nnH6**
>
> ---
> > `Weakness 3` More recent works about dynamic graphs should be properly included in related works or compared as baselines.
>
> Your advice is truly constructive! To address your concerns, we have incorporated two advanced DyGNN backbones, Roland [1] and SLATE [2], with their details summarized below:
>
> _Table C. DyGNN backbone statistics._
> |Baseline|GRCN|DySAT|SEIGN|Roland|SLATE|
> |-|-|-|-|-|-|
> |Venue|ICONIP|ICLR|ICDE|**KDD**|**SLATE**|
> |Year|2018|2020|2023|**2022**|**2024**|
>
> The experimental results are presented in Table D. We observe that DyAug maintains excellent generalizability across these backbones, achieving a 1.0% and 1.87% performance gain on Roland and SLATE, respectively.
>
> _Table D. Results on UCI dataset. Roland is implemented with Roland-GRU._
> |Backbone|Roland||SLATE||
> |-|-|-|-|-|
> | Vanilla       | 79.53±0.6122        |-                  | 81.26±0.3051        |-                  |
> | +DropEdge     | 79.25±0.4822        | -0.28 ↓          | 80.12±0.4715        | -1.14 ↓          |
> | +DropNode     | 79.29±0.6590        | -0.24 ↓          | 80.59±0.3552        | -0.67 ↓          |
> | +DropMessage  | 79.72±0.3145        | +0.19 ↑          | 81.77±0.5819        | +0.51 ↑          |
> | +GraphMixup   | 77.50±0.8911        | -2.03 ↓          | 80.35±0.7401        | -0.91 ↓          |
> | +NeuralSparse | 79.86±0.4818        | +0.33 ↑          | 81.75±0.4291        | +0.49 ↑          |
> | +SUBLIME      | 80.23±0.5741        | +0.70 ↑          | 82.56±0.3069        | +1.30 ↑          |
> | +RGDA         | 79.86±0.9582        | +0.33 ↑          | 82.77±0.3593        | +1.51 ↑          |
> | +DyAug        | **80.53±0.7318**    | +1.00 ↑          | **83.13±0.2639**    | +1.87 ↑          |
>
> ---
> > `Question 1` The source code repo is expired.
>
> We sincerely apologize for the oversight in our settings, which caused the anonymous GitHub repository to expire. We have now restored the anonymous link and ensured its accessibility. Thank you for your understanding!
>
> ---
> > `Question 2` In section 3.4, the description of $\overline{M} = 1_N - M$ and the specification that $1_N\in\{1\}^{N\times N}$ is an all-one matrix is somewhat perplexing. Given the typical sparsity of graphs, the transformation $\overline{M} = 1_N - M$ would result in a very dense matrix, thereby significantly increasing computation as the number of edges grows. This raises the question of whether the intended expression was $\overline{M} = A - M$, where represents the adjacency matrix of the graph, rather than an all-one matrix.
>
> Sorry for any confusion caused! You are absolutely correct—this was indeed a typo. We have now updated the corresponding equations throughout the manuscript to ensure consistency, replacing the term with **$\overline{M} = A - M$**. Thank you once again for your thorough review, which has significantly enhanced the quality of our work!
>
>
> ---
>
> [1] ROLAND: graph learning framework for dynamic graphs, KDD 2022
>
> [2] Supra-Laplacian Encoding for Transformer on Dynamic Graphs, NeurIPS 2024

---

> ### Author Response · Authors · 2024-11-26
> **Manuscript Revision Deadline Approaching; We Are Sincerely At Your Service!**
>
> Dear Reviewer nnH6,
>
> We would like to extend our sincere gratitude for your insightful review and dedication to the review process. Your invaluable feedback has significantly enhanced the quality of our manuscript. Given the approaching deadline for submitting the revised version, we are now providing the updated manuscript with highlighted changes for your reference.
>
> In particular, we have addressed your concerns as follows:
>
> 1. Conducted additional ablation studies across a broader range of scenarios (addressing `Weakness 1`).
> 2. Revised the complexity analysis and updated related formulas (addressing `Weakness 2` and `Question 2`).
> 3. Incorporated updated dynamic GNN models as backbones (addressing `Weakness 3`).
> 4. Maintained the source code repository (addressing `Question 1`).
>
> **We must express our genuine admiration for your exceptional academic intuition, which has guided our revisions.** Additionally, we remain fully committed to addressing any further concerns or questions you may have. We deeply value this opportunity for refinement and are eager to ensure the manuscript meets higher standards.
>
> Thank you once again for your indispensable support.
>
> Warm regards,
>
> Authors

---

> ### Comment · Reviewer_nnH6 · 2024-11-27
> **ACK**
>
> Thank you for the detailed rebuttal. My concerns have been addressed.

---

> > ### Author Response · Authors · 2024-11-27
> > **Thank you!**
> >
> > We truly appreciate your constructive feedback and continuous support, which we believe has further improved our work. Thank you again for your time and consideration!

---

### Official Review · Reviewer_7aTN · 2024-10-29

**Soundness:** 2
**Presentation:** 3
**Contribution:** 2
**Rating:** 5
**Confidence:** 4

**Summary:**

This paper introduces a novel framework for dynamic graph data augmentation called DyAug, addressing a critical challenge in applying static GDA methods to dynamic graphs: maintaining temporal consistency across graph snapshots. By combining causal subgraph rationalization with environmental subgraph augmentation, DyAug performs augmentation across three dimensions: spatial, temporal, and spatiotemporal. Extensive experimental results demonstrate the effectiveness of DyAug.

**Strengths:**

1. This paper tackles the challenge of maintaining temporal consistency across graph snapshots, an issue that, according to the authors, has not been previously addressed. The approach of combining causal subgraph rationalization with environmental subgraph augmentation to enhance the graph from spatial, temporal, and spatiotemporal perspectives is some what novel.
2. The paper demonstrates the performance, robustness, and generalization capacity of DyAug through comprehensive experiments.
3. The paper is generally clear, with a structured presentation that facilitates understanding of the DyAug. The figures are particularly effective in illustrating key concepts, such as the causal and the proposed augmentation strategies.
4. The significance of DyAug lies in its potential impact on the field of robustness dynamic GNNs.

**Weaknesses:**

1. The authors claim to be pioneers in exploring graph data augmentation (GDA) within dynamic graphs. However, several works on out-of-distribution (OOD) handling in dynamic graphs [1][2][3] have already incorporated augmentation strategies specifically for dynamic graphs. These works focus on enhancing the generalization ability of dynamic GNNs.
2. In DyAug, a causal identification mask is constructed to separate rational and environmental factors, followed by disturbing the spurious factors to augment the dynamic graph, ensuring that the GNN learns the causal factors. However, this approach aligns closely with standard techniques in dynamic graph OOD works [1][2][3], which also focus on dynamic graphs.
3. The paper argues that prior static graph augmentation methods overlook temporal consistency. However, it remains unclear how DyAug’s dynamic augmentation maintains this consistency. The authors only provide experimental evidence in Section 4.2 (Observation 1) to support temporal consistency. A more detailed explanation is needed to clarify which DyAug module specifically ensures that augmentation does not disrupt temporal consistency.
[1] Dynamic Graph Neural Networks Under Spatio-Temporal Distribution Shift, NeurIPS 2022
[2] Environment-aware dynamic graph learning for out-of-distribution generalization, NeurIPS 2024
[3]Spectral Invariant Learning for Dynamic Graphs under Distribution Shifts, NeurIPS 2024

**Questions:**

1. What are the limitations of dynamic graph augmentation methods targeting dynamic graph OOD, and what is the motivation to design DyAug under these previous methods?
2. How does DyAug's approach to causal discovery and causal perturbation differ from that of existing dynamic graph OOD methods? A more detailed explanation is needed.
3. Which component of DyAug ensures that its augmentation does not disrupt the temporal consistency of dynamic graphs, and why? This requires further clarification.

For details, please refer to the Weaknesses.

---

> ### Author Response · Authors · 2024-11-20
> **[Part 1/2] Response to Reviewer 7aTN**
>
> We sincerely thank you for your careful comments and thorough understanding of our paper! Here we give point-by-point responses to your comments and describe the revisions we made to address them.
>
> ---
> > `Weakness 1` The authors claim to be pioneers in exploring graph data augmentation (GDA) within dynamic graphs. However, several works on out-of-distribution (OOD) handling in dynamic graphs [1][2][3] have already incorporated augmentation strategies specifically for dynamic graphs. These works focus on enhancing the generalization ability of dynamic GNNs.
>
> Thank you for your profound insights, which have significantly enhanced the comprehensiveness of our manuscript! To address your concerns, we have made the following updates:
>
> **(1) Adding citations for [2,3]** We have already included DIDA [1] as a key baseline in Table 2. Additionally, we have incorporated relevant discussions about [2,3] in the updated manuscript.
>
>
> **(2) Clarifying distinctions between DyAug and existing dynamic graph OOD methods:** The following comparison highlights key differences:
>
> |Method|plug-in| temporal consistency | explicit causal subgraph separation |Spatial Aug.|Temporal Aug.|Spatial-temporal Aug.|Complexity|
> |-|-|-|-|-|-|-|-|
> |DIDA [1]|×|×|√|×|×|√| $\mathcal{O}(\|\mathcal{E}\| + \|\mathcal{V}\|d^2 + \|\mathcal{E}_p\|\|S\|d)$|
> |EAGLE [2]|×|×|√|×|×|√|$\mathcal{O}(\|\mathcal{E}\| \sum_l d^{(l)} +\mathcal{V}(\sum_l d^{(l-1)}d^{(l)} + (d^{(L)})^2 ) )$
> |SILD [3]|×|×|×|√|×|×| $\mathcal{O}(\|\mathcal{E}\|d+\|\mathcal{V}\|d + \|\mathcal{V}\|d^2 + \|\mathcal{V}\|d\log T + \|N_p\|Sd)$
> |DyAug (ours)|√|√|√|√|√|√|$\mathcal{O}\left( \|\mathcal{E}\|d + \|\mathcal{V}\|dT^2\right)$|
>
> From this comparison, we identify **three key differences** that underscore DyAug's contributions:
>
> 1. **Plug-and-play functionality:**
>    Methods such as [1,2,3] are designed for OOD-specific backbones, while DyAug can seamlessly integrate with any DyGNN to enhance performance, robustness, and generalizability.
> 2. **Temporal-consistency awareness:**
>    While [1,2,3] primarily rely on disentangled learning to generate multiple representations for a central node, DyAug takes a distinct approach by sequentially identifying causal masks for each graph snapshot, each conditioned on the previous one (as shown in Eq. (3)). This Markov-style subgraph identification ensures rationale consistency along the temporal dimension.
> 3. **Comprehensive augmentation:**
>    DIDA and EAGLE employ spatial-temporal augmentation by incorporating environment embeddings from other nodes across different timestamps. SILD, on the other hand, focuses on spatial augmentation by enhancing nodes using variant spectrum information from other nodes. DyAug stands out by providing **comprehensive augmentation across three dimensions: spatial, temporal, and spatial-temporal**.
>
>
> We hope these updates effectively showcase the novel contributions of DyAug.
>
>
>
> ---
> > `Weakness 2` In DyAug, a causal identification mask is constructed to separate rational and environmental factors, followed by disturbing the spurious factors to augment the dynamic graph, ensuring that the GNN learns the causal factors. However, this approach aligns closely with standard techniques in dynamic graph OOD works [1][2][3], which also focus on dynamic graphs.
>
> In response to `Weakness 1`, we have concluded the fundamental differences between our DyAug and existing dynamic graph OOD works. We sincerely hope this can address your concerns.
>
> ---
> > `Weakness 3` The paper argues that prior static graph augmentation methods overlook temporal consistency. However, it remains unclear how DyAug’s dynamic augmentation maintains this consistency.
>
> Your insightful comments are invaluable in highlighting the true contributions of DyAug! We would like to respectfully clarify that **Section 3 of the Methodology** introduces at least two dedicated designs aimed at preserving **temporal consistency**：
>
> - **Temporal-Conditioned Graph Rationalization (Line 266):**
>    During the generation of the causal mask $\mathbf{M}_t^\mathcal{R}$ for a graph snapshot $\mathcal{G}_t$, the process is explicitly conditioned on the causal subgraph of the $(t-1)$-th timestamp. This temporal dependency naturally mirrors the evolutionary characteristics of dynamic graphs over time and ensures temporal consistency across the rationales of consecutive timestamps.
> - **Consistency Regularization Loss (Eq. (6):**
>    We treat closely situated subsequences as positive pairs and those with greater temporal separation as negative pairs. The proposed loss encourages higher consistency within positive pairs, reinforcing temporal coherence across neighboring snapshots.
>
> These components, novel to the DyGNN / GDA literature, enable DyAug to robustly safeguard temporal consistency while achieving improved rationale discovery. We hope this clarification effectively addresses your concerns!

---

> ### Author Response · Authors · 2024-11-20
> **[Part 2/2] Response to Reviewer 7aTN**
>
> > `Question 1` What are the limitations of dynamic graph augmentation methods targeting dynamic graph OOD, and what is the motivation to design DyAug under these previous methods?
>
> Thank you for your insightful comment—we are honored to address your question! Please allow us to summarize the **limitations of existing dynamic graph OOD methods** and the corresponding **motivations behind DyAug**:
>
> - **Lack of Plug-and-Play Capability**
>     - **Limitation**: The ultimate goal of designing a graph data augmentation (GDA) method is to benefit all relevant backbones and datasets. However, existing DyGNN OOD methods are often tied to specific backbones, leaving their plug-and-play potential unexplored.
>     - **Motivation**: From its inception, DyAug has been designed as a universally applicable plugin, empowering any (DTDG) DyGNN without backbone restrictions.
> - **Unawareness of Temporal Consistency**
>     - **Limitation**: As discussed in Section 1, current GDA methods fail to preserve consistency across graph snapshots. Similarly, existing DyGNN OOD methods lack designs to address this issue. For instance, methods like DIDA and EAGLE employ disentangled learning for causal and non-causal separation, but this process is performed independently for each snapshot.
>     - **Motivation**: DyAug is specifically designed to ensure temporal consistency across snapshots. Detailed designs and explanations can be found in our response to `Weakness 3`.
> - **Limited Augmentation Dimensions**
>     - **Limitation**: Current DyGNN OOD methods overlook the varying augmentation needs of different datasets. Some datasets may prioritize temporal augmentation, while others may benefit more from spatial augmentation, as validated in our ablation study in `Appendix B.2`. However, DIDA and EAGLE focus solely on spatial-temporal interventions, and SILD considers only spatial interventions.
>     - **Motivation**: DyAug adopts a simple yet comprehensive design to simultaneously support **spatial**, **temporal**, and **spatial-temporal** augmentations.
> - **Lack of Robustness Measures**
>     - **Limitation**: The performance of existing DyGNN OOD methods under adversarial attacks remains unexplored.
>     - **Motivation**: DyAug is positioned as a **robustifier for DyGNNs**, effectively defending against structural, feature, and hybrid adversarial attacks.
>
> We sincerely hope this summary effectively highlights DyAug’s motivations and its unique contributions!
>
> ---
> > `Question 2` How does DyAug's approach to causal discovery and causal perturbation differ from that of existing dynamic graph OOD methods? A more detailed explanation is needed.
>
> We deeply admire your wisdom and expertise! In response to **`Weakness 1`**, we have conducted a detailed comparison between DyAug and existing dynamic graph OOD methods. We hope this analysis addresses your concerns effectively.
>
> ---
> > `Question 3` Which component of DyAug ensures that its augmentation does not disrupt the temporal consistency of dynamic graphs, and why? This requires further clarification.
>
> In response to **`Weakness 3`**, we have provided a detailed explanation of DyAug’s two dedicated mechanisms for maintaining temporal consistency. We are also more than happy to address any additional questions you may have!
>
> ---
> [1] Dynamic Graph Neural Networks Under Spatio-Temporal Distribution Shift, NeurIPS 2022
>
> [2] Environment-aware dynamic graph learning for out-of-distribution generalization, NeurIPS 2023
>
> [3] Spectral Invariant Learning for Dynamic Graphs under Distribution Shifts, NeurIPS 2023

---

> ### Author Response · Authors · 2024-11-26
> **Manuscript Revision Deadline Approaching; We Are Sincerely At Your Service!**
>
> Dear Reviewer 7aTN,
>
> We would like to thank you for your insightful review and dedication to the review process. With your help, the quality of the manuscript has been significantly improved. Due to the approaching deadline for submitting the revised manuscript, we are now submitting a revised version with highlighted changes.
>
> **Also, we would like to respectfully express our readiness to address any further concerns you may have.** At this stage, we have provided detailed responses regarding the similarities and differences between DyAug and current dynamic graph OOD methods, as well as DyAug's dedicated design for ensuring temporal consistency. We are eager to make the most of this revision opportunity, as we deeply value your invaluable insights.
>
> Thank you once again for your generous support.
>
> Warm regards,
>
> Authors

---

> ### Author Response · Authors · 2024-11-29
> **Gentle reminder to Reviewer 7aTN**
>
> Dear Reviewer 7aTN,
>
> This is a gentle reminder that **the discussion phase will end in less than 4 days**, but we have not yet received your feedback on our rebuttal. We respectfully understand that due to your other important commitments, your time is precious and the demand is high. But we are eager to collaborate with you to improve this paper, and we have made extensive contributions and efforts to this end. We sincerely hope that you find our response convincing and kindly consider revisiting your rating.
>
> We would like to express our gratitude once again to the reviewer for their time and all constructive feedback provided!
>
> Thanks and Regrads,
>
> Submission 4108 authors

---

> > ### Comment · Reviewer_7aTN · 2024-11-29
> >
> > Thanks for the response. Even though the authors discuss the differences from previous work on plug and play and temporal consistency, the novelty of this paper is still my major concern. So I decided to keep my rating.

---

> > > ### Author Response · Authors · 2024-11-30
> > >
> > > We sincerely appreciate your recognition of our rebuttal, especially regarding our dynamic graph GDA method, DyAug, and its **significant differences** from dynamic graph OOD methods in terms of **(1) plug-and-play functionality and (2) temporal consistency modeling, which are the core contributions of our work**. We are also delighted to realize that we have successfully clarified DyAug's two unique designs for addressing temporal consistency. We are more than willing to address any further specific concerns you may have. Thank you once more!

---

### Official Review · Reviewer_uP12 · 2024-11-02

**Soundness:** 3
**Presentation:** 3
**Contribution:** 3
**Rating:** 6
**Confidence:** 4

**Summary:**

This paper introduces DyAug, a novel framework designed to enhance dynamic graph neural networks (DyGNNs) through graph rationalization and data augmentation, addressing the limitations of static graph augmentation methods when applied to dynamic graphs. DyAug utilizes temporal consistency-aware augmentation strategies to separate causal and non-causal graph components across time, leading to improved performance, generalization, and robustness against adversarial attacks and temporal distribution shifts. Extensive experiments demonstrate that DyAug consistently outperforms state-of-the-art methods.

**Strengths:**

1. The proposed DyAug framework extends GDA to dynamic graphs, a less explored area. It offers an innovative approach by focusing on maintaining temporal consistency, which is crucial for dynamic graphs.
2. DyAug effectively maintains temporal consistency, ensuring that augmentations do not disrupt the natural evolution of dynamic graphs.
3. DyAug introduces the concept of graph rationalization for dynamic GNNs, separating causal subgraphs from non-causal parts. This enables the model to learn more meaningful temporal representations.

**Weaknesses:**

1. DyAug introduces additional computational complexity from causal mask estimation, contrastive loss, and consistency loss, which could pose scalability issues for large graphs.
2. While the paper includes comparisons with several GDA methods, some notable methods like DIR [1] and GREA [2] were excluded due to compatibility issues. This might make the performance comparisons less comprehensive.
3. DyAug's augmentation techniques (spatial, temporal, spatial-temporal replacements) rely on heuristic selection strategies, which might not generalize well across all types of dynamic graphs.
4. Although an ablation study is provided, it could be more exhaustive, especially in terms of testing the impact of different hyperparameters and rationale generation methods on diverse datasets.

[1] Discovering invariant rationales for graph neural networks.
[2] Graph rationalization with environment-based augmentations.

**Questions:**

1. How does DyAug perform on large-scale dynamic graphs with millions of nodes and edges, especially considering the additional complexity introduced by rationalization and consistency regularization?
2. The current focus is on discrete-time dynamic graphs. How challenging would it be to adapt DyAug to continuous-time dynamic graphs (CTDGs)?
3. Could there be scenarios where the rationale-environment separation introduces new spurious correlations, and how can this be mitigated?
4. How sensitive is the model's performance to the choice of window size $w$ in the consistency regularization loss, and does this affect the robustness of DyAug?

---

> ### Author Response · Authors · 2024-11-20
> **[Part 1/3] Response to Reviewer uP12**
>
> > `Weakness 1` DyAug introduces additional computational complexity from causal mask estimation, contrastive loss, and consistency loss, which could pose scalability issues for large graphs.
>
> Your feedback is crucial! To address your concerns, we first (1) review the complexity of DyAug, and (2) test DyAug on larger-scale graphs.
>
> **Complexity of DyAug** As analyzed in Section 3.6, the computational complexity introduced by DyAug is $\mathcal{O}\left(\sum_{t=1}^T |\mathcal{E}^{(t)}|D + NDT^2\right)$, which is smaller than the complexity of most DyGNNs, e.g., $\mathcal{O}(N \cdot D^2 \cdot T + N \cdot D \cdot T^2)$ for DySAT. The key question, then, is whether this complexity restricts DyAug's scalability to larger graphs. In the following, we empirically examine this concern.
>
> **DyAug on larger-scale graphs** We extended our experiments to include a large-scale dynamic graph dataset, DGraphFin [2], using the slicing and data splitting strategy outlined in [1]. This dataset comprises 4,889,537 nodes and 4,300,999 edges, as detailed in Table A. We believe this benchmark effectively tests DyAug's scalability.
>
>
>
> _Table A. Dataset characteristics._
> |Dataset|COLLAB|Yelp|Bitcoin|UCI|ACT|DGraphFin (new)|
> |-|-|-|-|-|-|-|
> |#Nodes|23,035|13,095|5,881|1,899|20,408|4,889,537|
> |#Edges|151,790|65,375|35,591|59,835|202,339|4,300,999
> |#Time steps|16|24|21|13|30|12|
>
>
> We evaluated DyAug's performance on DGraphFin using DySAT as the backbone model, as shown in Table B.
>
> _Table B. Results on DGraphFin+DySAT. The imeplementation DySAT is from Amazon-TGL (https://github.com/amazon-science/tgl). The results are reported on a single NVDIA Tesla A100 40G GPU._
> |Metric|AUC (%)|Per-epoch time (s)|GPU Memory (GB)|
> |-|-|-|-|
> |Vanilla|73.26|47s|3.7GB|
> |+DropEdge|72.69|43s|3.7GB|
> |+DropNode|73.50|41s|3.7GB|
> |+DropMessage|71.22|53s|3.7GB|
> |+NeuralSparse|70.80|186s|17GB|
> |+SUBLIME|73.15|147s|21GB|
> |+RGDA|74.12|69s|6.9GB|
> |+DyAug|**74.98**|56s|5.6GB|
>
> From Table B, we can conclude that DyAug seamlessly scales to ultra-large graphs. Graph structure learning methods like NeuralSparse and SUBLIME impose substantial GPU memory demands with marginal or even negative performance gains ($2.46\%\downarrow$ for NeuralSparse). Rule-based augmenters such as DropEdge and DropNode introduce negligible computational overhead but fail to deliver notable performance gains, consistent with the findings in Table 1. DyAug, however, achieves a $1.72\%$ AUC improvement with an additional GPU memory burden of less than 2 GB, demonstrating its exceptional scalability.
>
>
> ---
> > `Weakness 2` While the paper includes comparisons with several GDA methods, some notable methods like DIR [1] and GREA [2] were excluded due to compatibility issues. This might make the performance comparisons less comprehensive.
>
> Thank you for your comment. Allow us to address your concerns from the following two perspectives:
>
> **DIR and GREA are graph rationalization methods specifically designed for graph classification tasks, which are generally not considered in DyGNN methods**. As explicitly stated in DIR and GREA, these methods are tailored for graph-level tasks and typically involve $O(N^2)$ complexity (Eq. (5) in DIR). When adapted to dynamic graphs, this results in $O(T \cdot N^2)$ complexity, which is computationally prohibitive for most DyGNNs. For example, applying DIR to Yelp+DySAT would increase the GPU memory requirement by an additional 88GB. Other classic DyGNN works, such as [3], have also excluded DIR and GREA, making it clear that this is a common practice within the research community and does not limit the comprehensiveness of our work.
>
> **We include the compatible graph rationalization method RGDA as our baseline**. While DIR and GREA are incompatible, we have selected RGDA, to the best of our knowledge, the only graph rationalization method that is compatible with dynamic graphs, as our baseline. The results in Table 1 clearly demonstrate that DyAug consistently outperforms RGDA, highlighting the superiority of our method.

---

> ### Author Response · Authors · 2024-11-20
> **[Part 2/3] Response to Reviewer uP12**
>
> > `Weakness 3`: DyAug's augmentation techniques (spatial, temporal, spatial-temporal replacements) rely on heuristic selection strategies, which might not generalize well across all types of dynamic graphs.
>
> Thank you for highlighting this issue! Although the current selection strategy is heuristic, we respectfully propose that DyAug can be enhanced with a more generalizable augmentation selection strategy through a minimal code modification of **no more than 20 lines**. Specifically, we introduce an augmentation router, $\Psi$, inspired by mixture-of-experts (MoE) [4], which adaptively determines the appropriate augmentation strategy for each node:
>
> $$
> \Psi(\mathbf{h}\_{t,i}^\mathcal{R})=\operatorname{Softmax}(\operatorname{TopK}(\psi(\mathbf{h}\_{t,i}^\mathcal{R}), k)), \quad \psi(\mathbf{h}\_{t,i}^\mathcal{R}) = \mathbf{h}\_{t,i}^\mathcal{R}W_r
> $$
>
> where $k=1$, $W_r \in \mathbb{R}^{4 \times D}$ is a trainable parameter, and $\psi(\mathbf{h}_{t,i}^\mathcal{R}) \in \mathbb{R}^4$ represents the computed scores for the four augmentation strategies (no augmentation, spatial, temporal, and spatial-temporal). Based on the highest score computed by $\Psi(\cdot)$, we assign each node its corresponding GDA strategy, thereby improving the data adaptability and generalizability of DyAug.
>
> **Experimental Evaluation** To assess whether DyAug with $\Psi$ offers better generalizability, we report its performance on the COLLAB and Yelp datasets, as shown in Table C. It is evident that DyAug with $\Psi$ not only matches DyAug's performance under standard settings, but also outperforms it by $2.07\%$ in the presence of distribution shifts. This improvement is attributed to DyAug with $\Psi$'s enhanced ability to adaptively select the augmentation strategy for central nodes.
>
> _Table C. AUC score (%) of different methods with and without distribution shift (DS) on GRCN backbone._
> |Dataset|COLLAB||YELP||
> |-|-|-|-|-|
> |Test data|w/o DS|w/ DS|w/o DS|w/ DS|
> |Vanilla|82.78±0.52|67.49±0.73|66.45±1.87| 61.82±3.39
> |+DyAug|**84.95±0.70**| 73.22±0.56 |67.95±0.24| 61.78±2.76
> |+DyAug w/ $\Psi$|84.72±0.89|**75.39±0.82**|**68.20±0.74**|**63.85±3.15**
>
> ---
> > `Weakness 4` Although an ablation study is provided, it could be more exhaustive, especially in terms of testing the impact of different hyperparameters and rationale generation methods on diverse datasets.
>
> Thank you for your valuable suggestion! To provide a more comprehensive evaluation of DyAug, we have included a sensitivity analysis on the window size $\varpi$ using the COLLAB and Yelp datasets, as shown in Table D:
>
> _Table D. Sensitivity anlysis on parameter $\varpi$ on GRCN backbone._
> |Dataset|COLLAB|Yelp|
> |-|-|-|
> |$\varpi=2$|0.8495±0.0070|0.6795±0.0204|
> |$\varpi=4$|0.8506±0.0082|0.6785±0.0139|
> |$\varpi=6$|0.8469±0.0069|0.6732±0.0188|
> |$\varpi=8$|0.8345±0.0112|0.6690±0.0228|
>
>
> It can be observed that DyAug performs consistently well when $\varpi \in \{2,4\}$, while for $\varpi = 8$, there is a notable performance drop on the COLLAB dataset ($1.50\% \downarrow$ compared to $\varpi = 2$). We attribute this to the overly large window size, as the COLLAB dataset consists of only 16 snapshots. With such a large window, the majority of the dynamic graph sequence is considered as "positive pairs," rendering $\mathcal{L}_{cr}$ ineffective. Therefore, we have fixed $\varpi = 2$ for all subsequent experiments.

---

> ### Author Response · Authors · 2024-11-20
> **[Part 3/3] Response to Reviewer uP12**
>
> > `Question 1` How does DyAug perform on large-scale dynamic graphs with millions of nodes and edges, especially considering the additional complexity introduced by rationalization and consistency regularization?
>
> In response to `Weakness 1`, we evaluated DyAug on DGraphFin, which contains 4,889,537 nodes and 4,300,999 edges. We hope this serves as additional validation of DyAug's scalability.
>
>
> ---
> > `Question 2` The current focus is on discrete-time dynamic graphs. How challenging would it be to adapt DyAug to continuous-time dynamic graphs (CTDGs)?
>
> To address this question, allow us to review the current methods for modeling and augmenting dynamic graphs for CTDGs and DTDGs:
>
> **For dynamic graph modeling**:
> - Classical models such as JODIE, TGAT, and DyRep are specifically designed for CTDGs.
> - In contrast, methods like EvolveGCN, Roland, DySAT, and SEIGN are tailored for DTDGs.
>
> **For dynamic graph augmentation**:
> - For CTDGs, specialized GDA methods [5,6] are exclusively applicable to CTDGs and their corresponding DyGNNs.
> - For DTDGs, our method, DyAug, is heavily reliant on the graph snapshot sequence data format.
>
> As shown, the research on CTDGs and DTDGs is relatively orthogonal, each presenting unique challenges and methodologies. Therefore, we respectfully note that **although DyAug is currently not applicable to CTDGs, this does not diminish its contribution**. DTDGs are crucial for scenarios where edge timestamps are inherently discrete (e.g., daily price-based stock correlation graphs) or when fine-grained edge timestamps are unavailable. This representation supports over 40 DyGNN backbones and 15 datasets, all of which DyAug integrates with seamlessly. **Thus, we believe that DTDG GDA methods remain an important and meaningful area of exploration.**
>
>
> ---
> > `Question 3` Could there be scenarios where the rationale-environment separation introduces new spurious correlations, and how can this be mitigated?
>
> Thank you for your insightful comments! We fully acknowledge that rationale-environment separation is inherently (or inevitably) imperfect and may introduce new spurious correlations—an issue faced by all graph rationalization methods. As depicted in Figure 3, DyAug aims to decompose $\mathcal{G}\_{1:T}$ into the causal factor $\mathcal{C}$ and the non-causal factor $\mathcal{S}$. However, the result is often an approximation, $\mathcal{C}'$ and $\mathcal{S}'$, where both $\mathcal{C} \setminus \mathcal{C}'$ and $\mathcal{S} \setminus \mathcal{S}'$ represent modeling errors. These errors could potentially lead to new spurious correlations, such as $\mathcal{C} \leftarrow \mathcal{G}\_{1:T} \rightarrow \mathcal{S} \setminus \mathcal{S}' \rightarrow \mathbf{A}\_{1:T} \rightarrow \mathcal{H} \rightarrow \mathcal{Y}$. However, we respectfully emphasize the following points:
>
> 1. The negative impact of these spurious correlations is significantly smaller than that of the original ones. This is because $\mathcal{S} \setminus \mathcal{S}' \ll \mathcal{S}$, resulting in much lower path flow and, consequently, a substantially reduced confounding bias.
> 2. This minor noise is effectively mitigated by our multi-dimensional augmentation mechanism (spatial, temporal, and spatial-temporal), which enhances the model’s robustness.
>
> ---
> > `Question 4` How sensitive is the model's performance to the choice of window size in the consistency regularization loss, and does this affect the robustness of DyAug?
>
> Thank you for your comment! In response to `Weakness 4`, we have conducted a sensitivity analysis of the window size $\varpi$ on two datasets. We hope this additional evaluation adequately addresses your concerns.
>
>
> ---
> [1] A Comprehensive Survey of Dynamic Graph Neural Networks: Models, Frameworks, Benchmarks, Experiments and Challenges
>
> [2] The DGraphFin datasets
>
> [3] Dynamic Graph Neural Networks Under Spatio-Temporal Distribution Shift, NeurIPS 2022
>
> [4] Outrageously large neural networks: The sparsely-gated mixture-of-experts layer, ICLR 2017
>
> [5] Adaptive data augmentation on temporal graphs, NeurIPS 2021
>
> [6] Temporal graph representation learning with adaptive augmentation contrastive, ECML-PKDD 2023

---

> ### Author Response · Authors · 2024-11-26
> **Manuscript Revision Deadline Approaching; We Are Sincerely At Your Service!**
>
> Dear Reviewer uP12,
>
> We would like to thank you for your insightful review and dedication to the review process. With your help, the quality of the manuscript has been significantly improved. Due to the approaching deadline for submitting the revised manuscript, we are now submitting a revised version with highlighted changes.
>
> **Also, we would like to respectfully convey our willingness to address any further concerns you may have.** We aim to make the most of this opportunity for revision, as we deeply value your invaluable insights.
>
> Thank you once again for your priceless support.
>
> Warm regards,
>
> Authors

---

> > ### Comment · Reviewer_uP12 · 2024-11-27
> >
> > Thanks for your response. Some of my concerns have been addressed. I will keep my positive rating.

---

> > > ### Author Response · Authors · 2024-11-27
> > > **Thank you!**
> > >
> > > We are more than glad to hear that our rebuttal has effectively addressed your concerns! Thank you again for your constructive and thoughtful comments!

---

### Official Review · Reviewer_AF5m · 2024-11-04

**Soundness:** 3
**Presentation:** 2
**Contribution:** 3
**Rating:** 6
**Confidence:** 4

**Summary:**

In this paper, the authors identify limitations in applying existing graph data augmentation (GDA) techniques to dynamic graphs. Through a detailed investigation, they reveal that these methods often degrade performance by disrupting temporal consistency within the graph structures. To address this challenge, the authors introduce DyAug, a graph data augmentation approach specifically designed for dynamic graphs. DyAug achieves this by partitioning the input graph into a "rational" component, which preserves essential temporal consistency, and an "environmental" component, which can be modified to enrich the training data. Building on this partitioning, they propose three replacement strategies to selectively augment the dynamic graph without compromising consistency. Experimental results demonstrate that DyAug significantly improves both the performance and robustness of dynamic GNNs.

**Strengths:**

The paper has the following strengths:

- The authors focus on emerging dynamic graphs, a format increasingly common in real-world applications, making the approach highly relevant.

- The proposed approach is well-founded and effectively enhances the resulting model's performance and robustness.

- The authors thoroughly compare various existing GDA methods, demonstrating significant improvements in outcomes.

**Weaknesses:**

The paper has the following weaknesses:

- The proposed scheme's effectiveness is not evaluated on continuous-time dynamic graphs (CTDGs), which are also crucial in dynamic graph learning.

- The scalability of the approach is not thoroughly assessed, leaving uncertainty about its performance on larger graphs.

- The dynamic GNN models used in the evaluation are relatively outdated, potentially limiting the generalizability of the results.

- The adversarial attacks used for testing the approach are not as advanced as state-of-the-art methods, which may affect the robustness evaluation.

**Questions:**

Here are some questions that may help strengthen the overall merit of the paper:

1. Have you considered evaluating the proposed scheme on continuous-time dynamic graphs (CTDGs)? How do you anticipate the approach would perform on CTDGs compared to discrete-time settings?

2. Could you provide insights into the scalability of your approach? Have you tested it on larger graphs, and if so, how does performance scale with increasing graph size? Additionally, what are the associated overheads?

3. Could you share more evaluation results of the approach on recent models to provide a broader validation of its effectiveness? For instance, models such as TGAT [1], ROLAND [2], or DyGFormer [3]?

4. Given that the adversarial attacks used in the evaluation are not the most advanced, do you plan to test your approach against state-of-the-art adversarial attacks, such as [4]? How do you anticipate your method would perform in these more challenging scenarios?

[1] Xu, Da, et al. "Inductive representation learning on temporal graphs." arXiv preprint arXiv:2002.07962 (2020).

[2] You, Jiaxuan, Tianyu Du, and Jure Leskovec. "ROLAND: graph learning framework for dynamic graphs." Proceedings of the 28th ACM SIGKDD conference on knowledge discovery and data mining. 2022.

[3] Yu, Le, et al. "Towards better dynamic graph learning: New architecture and unified library." Advances in Neural Information Processing Systems 36 (2023): 67686-67700.

[4] Sharma, Kartik, et al. "Temporal dynamics-aware adversarial attacks on discrete-time dynamic graph models." Proceedings of the 29th ACM SIGKDD Conference on Knowledge Discovery and Data Mining. 2023.

---

> ### Author Response · Authors · 2024-11-20
> **[Part 1/3] Response to Reviewer AF5m**
>
> We sincerely thank Reviewer AF5m for the thoughtful and constructive reviews of our manuscript! Based on your questions and recommendations, we give point-by-point responses to your comments and describe the revisions we made to address them.
>
> ---
> > `Weakness 1`: The proposed scheme's effectiveness is not evaluated on continuous-time dynamic graphs (CTDGs), which are also crucial in dynamic graph learning.
>
> Thank you for your thoughtful comment! **As stated on Line 192, DyAug focuses on the graph data augmentation (GDA) for DTDGs.** DTDGs and CTDGs represent two primary paradigms of dynamic graphs, each with distinct challenges and methodologies. To date, no model can seamlessly operate across both representations, whether from the perspective of backbone models or GDA. For example:
> - Classical models such as JODIE, TGAT, and DyRep are exclusively designed for CTDGs.
> - Conversely, methods like EvolveGCN, Roland, and DySAT are tailored to DTDGs.
>
> Similarly, many graph data augmentation approaches depend heavily on the data format, making direct cross-format applications infeasible. **Nevertheless, we respectfully believe this does not diminish DyAug's contribution:** DTDGs, characterized by graph snapshots over discrete time intervals, are essential for scenarios where edge timestamps are inherently discrete (e.g., daily price-based stock correlation graphs) or where fine-grained edge timestamps are unavailable. This representation supports over 40 DyGNN backbones and 15 datasets, all of which DyAug can seamlessly integrate with.
>
> Nevertheless, we believe it is a promising direction to implement DyAug on CTDGs, and we have included this point in Section 5 of our updated manuscript.
>
>
> ---
> > `Weakness 2` The scalability of the approach is not thoroughly assessed, leaving uncertainty about its performance on larger graphs.
>
> Thank you for your insightful inquiry! To address your question, we will first (1) review the computational complexity of DyAug and then (2) report its performance on larger-scale graphs.
>
> **Complexity of DyAug** As analyzed in Section 3.6, the computational complexity introduced by DyAug is $\mathcal{O}\left(\sum_{t=1}^T |\mathcal{E}^{(t)}|D + NDT^2\right)$, which is smaller than the complexity of most DyGNNs, e.g., $\mathcal{O}(N \cdot D^2 \cdot T + N \cdot D \cdot T^2)$ for DySAT. The key question, then, is whether this complexity restricts DyAug's scalability to larger graphs. In the following, we empirically examine this concern.
>
> **DyAug on larger-scale graphs** We extended our experiments to include a large-scale dynamic graph dataset, DGraphFin [2], using the slicing and data splitting strategy outlined in [1]. This dataset comprises 4,889,537 nodes and 4,300,999 edges, as detailed in Table A. We believe this benchmark effectively tests DyAug's scalability.
>
>
>
> _Table A. Dataset characteristics._
> |Dataset|COLLAB|Yelp|Bitcoin|UCI|ACT|DGraphFin (new)|
> |-|-|-|-|-|-|-|
> |#Nodes|23,035|13,095|5,881|1,899|20,408|4,889,537|
> |#Edges|151,790|65,375|35,591|59,835|202,339|4,300,999
> |#Time steps|16|24|21|13|30|12|
>
>
> We evaluated DyAug's performance on DGraphFin using DySAT as the backbone model, as shown in Table B.
>
> _Table B. Results on DGraphFin+DySAT. The implementation of DySAT is from Amazon-TGL (https://github.com/amazon-science/tgl). The results are reported on a single NVIDIA Tesla A100 40G GPU._
> |Metric|AUC (%)|Per-epoch time (s)|GPU Memory (GB)|
> |-|-|-|-|
> |Vanilla|73.26|47s|3.7GB|
> |+DropEdge|72.69|43s|3.7GB|
> |+DropNode|73.50|41s|3.7GB|
> |+DropMessage|71.22|53s|3.7GB|
> |+NeuralSparse|70.80|186s|17GB|
> |+SUBLIME|73.15|147s|21GB|
> |+RGDA|74.12|69s|6.9GB|
> |+DyAug|**74.98**|56s|5.6GB|
>
> From Table B, we can conclude that DyAug seamlessly scales to ultra-large graphs. Graph structure learning methods like NeuralSparse and SUBLIME impose substantial GPU memory demands with marginal or even negative performance gains ($2.46\%\downarrow$ for NeuralSparse). Rule-based augmenters such as DropEdge and DropNode introduce negligible computational overhead but fail to deliver notable performance gains, consistent with the findings in Table 1. DyAug, however, achieves a $1.72\\%$ AUC improvement with an additional GPU memory burden of less than 2 GB, demonstrating its exceptional scalability.

---

> ### Author Response · Authors · 2024-11-20
> **[Part 2/3] Response to Reviewer AF5m**
>
> > `Weakness 3` The dynamic GNN models used in the evaluation are relatively outdated, potentially limiting the generalizability of the results.
>
> Thank you for your valuable feedback, which has further strengthened our manuscript! To address your concerns, we have incorporated two advanced DyGNN backbones, Roland [3] and SLATE [4], with their details summarized below:
>
> _Table C. DyGNN backbone statistics._
> |Baseline|GRCN|DySAT|SEIGN|Roland|SLATE|
> |-|-|-|-|-|-|
> |Venue|ICONIP|ICLR|ICDE|**KDD**|**NeurIPS**|
> |Year|2018|2020|2023|**2022**|**2024**|
>
> The experimental results are presented in Table D. We observe that DyAug maintains excellent generalizability across these backbones, achieving a 1.0% and 1.87% performance gain on Roland and SLATE, respectively.
>
> _Table D. Results on UCI dataset. Roland is implemented with Roland-GRU._
> |Backbone|Roland||SLATE||
> |-|-|-|-|-|
> | Vanilla       | 79.53±0.6122        |-                  | 81.26±0.3051        |-                  |
> | +DropEdge     | 79.25±0.4822        | -0.28 ↓          | 80.12±0.4715        | -1.14 ↓          |
> | +DropNode     | 79.29±0.6590        | -0.24 ↓          | 80.59±0.3552        | -0.67 ↓          |
> | +DropMessage  | 79.72±0.3145        | +0.19 ↑          | 81.77±0.5819        | +0.51 ↑          |
> | +GraphMixup   | 77.50±0.8911        | -2.03 ↓          | 80.35±0.7401        | -0.91 ↓          |
> | +NeuralSparse | 79.86±0.4818        | +0.33 ↑          | 81.75±0.4291        | +0.49 ↑          |
> | +SUBLIME      | 80.23±0.5741        | +0.70 ↑          | 82.56±0.3069        | +1.30 ↑          |
> | +RGDA         | 79.86±0.9582        | +0.33 ↑          | 82.77±0.3593        | +1.51 ↑          |
> | +DyAug        | **80.53±0.7318**    | +1.00 ↑          | **83.13±0.2639**    | +1.87 ↑          |
>
>
> ---
> > `Weakness 4` The adversarial attacks used for testing the approach are not as advanced as state-of-the-art methods, which may affect the robustness evaluation.
>
> Thank you for your valuable suggestion! Following your advice, we conducted additional experiments to evaluate DyAug under the more advanced attack method TD-PGD [7], as shown in Table E. The results demonstrate that DyAug maintains strong robustness even under TD-PGD attacks. Notably, at $\epsilon=0.1$, DyAug even achieves a performance improvement of $0.14\\%$ when applied on DySAT+UCI. What's more, DyAug demonstrated significant robustness on the DySAT+ACT setting under the TD-PGD attack, as it successfully improved the attacked performance from $68.53\\%$ to $74.90\\%$ at $\epsilon=0.3$, achieving a remarkable $6.37\\%$ gain, which we believe serves as compelling evidence of its resilience in more challenging scenarios.
>
>
> _Table E. Results tested with TP-PGD attack._
> |Dataset|Method|$\epsilon=0.0$|$\epsilon=0.1$|$\epsilon=0.2$|$\epsilon=0.3$|
> |-|-|-|-|-|-|
> |UCI|DySAT|0.7502±0.0056|0.7418±0.0092|0.7276±0.0143|0.7191±0.0129|
> ||DySAT+DyAug|0.7698±0.0063|0.7712±0.0080|0.7456±0.0120|0.7466±0.0174|
> |ACT|DySAT|0.7790±0.0036|0.7698±0.0028|0.7334±0.0138|0.6853±0.0262|
> ||DySAT+DyAug|0.7860±0.0054|0.7834±0.0112|0.7682±0.0109|0.7490±0.0176|
>
> ---
> > `Question 1` Have you considered evaluating the proposed scheme on continuous-time dynamic graphs (CTDGs)? How do you anticipate the approach would perform on CTDGs compared to discrete-time settings?
>
> We would like to clarify that DyAug is specifically designed as a GDA method tailored for DTDG data. However, applying DyAug to CTDGs is entirely feasible by converting CTDGs into graph snapshot sequences via time slicing. In fact, as mentioned in our response to `Weakness 2`, the DGraphFin dataset used was originally a CTDG. By adopting the transformation settings outlined in [1], we made it fully compatible with DyAug.
>
> Admittedly, DyAug is currently not applicable to the original data format of CTDGs. **Nonetheless, we believe this does not diminish the contributions of our work.** DTDGs and CTDGs represent two foundational paradigms of dynamic graphs, each tailored to distinct use cases. DTDGs, defined by graph snapshots over discrete time intervals, are indispensable for scenarios where edge timestamps are inherently discrete (e.g., daily price-based stock correlation graphs) or where fine-grained edge timestamps are unavailable. As we already discussed on Line 192, while existing augmentation techniques for CTDGs, such as those in [5,6], are specifically designed for CTDG DyGNNs, no comparable GDA method exists for DTDGs that seamlessly integrates with all DTDG-based DyGNNs. DyAug fills this gap by providing a generalizable and efficient solution for DTDG scenarios.

---

> ### Author Response · Authors · 2024-11-20
> **[Part 3/3] Response to Reviewer AF5m**
>
> > `Question 2` Could you provide insights into the scalability of your approach? Have you tested it on larger graphs, and if so, how does performance scale with increasing graph size? Additionally, what are the associated overheads?
>
> Thank you for your insightful comment! In our response to `Weakness 2`, we have provided additional performance details of DyAug on the large-scale graph DGraphFin, which contains 4,889,537 nodes. We have reported results on performance, per-epoch time and GPU memory usage, hoping to address your concerns.
>
> ---
> > `Question 3` Could you share more evaluation results of the approach on recent models to provide a broader validation of its effectiveness? For instance, models such as TGAT [1], ROLAND [2], or DyGFormer [3]?
>
> Thank you for your valuable suggestion! In our response to `Weakness 3`, we have included additional experiments demonstrating the integration of DyAug with two advanced DyGNN backbones, Roland (KDD 2022) and SLATE (NeurIPS 2024). We hope this further highlights the applicability of DyAug.
>
> ---
> > `Question 4` Given that the adversarial attacks used in the evaluation are not the most advanced, do you plan to test your approach against state-of-the-art adversarial attacks, such as [4]? How do you anticipate your method would perform in these more challenging scenarios?
>
> In response to `Weakness 4`, we conducted additional experiments to assess DyAug's robustness under the advanced TD-PGD attack. We hope this addresses your concerns and further substantiates DyAug's ability to maintain robustness in challenging scenarios.
>
> ---
>
> [1] A Comprehensive Survey of Dynamic Graph Neural Networks: Models, Frameworks, Benchmarks, Experiments and Challenges
>
> [2] The DGraphFin datasets
>
> [3] ROLAND: graph learning framework for dynamic graphs, KDD 2022
>
> [4] Supra-Laplacian Encoding for Transformer on Dynamic Graphs, NeurIPS 2024
>
> [5] Adaptive data augmentation on temporal graphs, NeurIPS 2021
>
> [6] Temporal graph representation learning with adaptive augmentation contrastive, ECML-PKDD 2023
>
> [7] Temporal Dynamics-Aware Adversarial Attacks on Discrete-Time Dynamic Graph Models, KDD2023

---

> ### Comment · Reviewer_AF5m · 2024-11-23
> **Concern Addressed**
>
> Thank you for the detailed experiments and clarifications. The responses have addressed my concerns. Therefore, I raise my score.

---

> > ### Author Response · Authors · 2024-11-24
> > **Thank you!**
> >
> > We extend our heartfelt thanks for your increased support of our paper! We are pleased that our rebuttal have sufficiently addressed your concerns.

---

### Author Response · Authors · 2024-11-20
**Summary of Manuscript Update**

Dear Reviewers,

Thank you immensely for your thoughtful and constructive comments! We are really encouraged to see that the reviewers appreciate some positive aspects of our paper, such as **an important problem** (Reviewers `AF5m`, `uP12`, `7aTN`), **well-organized presentation** (Reviewer `7aTN`) , **significant improvements** (Reviewers `AF5m`, `7aTN`, `nnH6`).

Your expertise significantly helps us strengthen our manuscript – this might be the most helpful review we have received in years! In addition to addressing your thoughtful comments point-to-point on the OpenReview forum, we have made the following modifications to the newly uploaded manuscript (all updated text is highlighted in blue):

* **Additional ablation study:** We have included ablation studies under distribution shift, feature attack, and Nettack scenarios in `Appendix B.2`.
* **Additional sensitivity analysis:** Sensitivity experiments for parameter $\varpi$ have been added in `Appendix B.3`.
* **Scalability on larger graphs:** Results for DyAug's performance on the large-scale DGraphFin dataset are now provided in `Appendix B.4`.
* **Additional attack methods:** We have evaluated DyAug's defense capabilities against TD-PGD attacks in `Appendix B.5`.
* **Combination with more DyGNN backbones:** We have evaluated DyAug with Roland and SLATE in `Appendix B.6`.
* **Other revisions:** Discussions related to dynamic graph OOD methods have been supplemented in `Appendix C`, and several typos have been corrected.

We have made earnest efforts to address the primary concerns raised. We also respectfully look forward to the thoughtful feedback from the reviewers to further enhance the quality of our manuscript.

Sincerely,

Authors

---

### Author Response · Authors · 2024-11-22
**General Response**

Dear Reviewers,

Thank you for your thorough and insightful reviews. We sincerely appreciate your valuable feedback, which has greatly improved our paper. Below, we summarize the primary concerns raised and our corresponding responses:

- **Scalability of DyAug** (`Reviewers AF5m, uP12`)
  We have included results demonstrating DyAug's performance on million-node DGraphFin dataset.
- **Experiments with additional DyGNN models** (`Reviewer AF5m`)
  We have reported experimental results combining DyAug with Roland (KDD 2022) and SLATE (NeurIPS 2024).
- **Defense against advanced adversarial attacks** (`Reviewer AF5m`)
  We have evaluated DyAug's robustness against the more advanced TD-PGD adversarial attack.
- **Additional ablation and sensitivity studies** (`Reviewers uP12, nnH6`)
  We have conducted and reported additional ablation studies across diverse scenarios and included a sensitivity analysis for the parameter $\varpi$.
- **Evaluation on CTDGs** (`Reviewers AF5m, uP12`)
  We have analyzed the distinctions between DyGNN backbones and GDA methods for CTDGs and DTDGs, underscoring their mutual incompatibility.
- **Comparison with dynamic graph OOD methods** (`Reviewer 7aTN`)
  We have performed a detailed comparison between DyAug and current dynamic graph OOD methods, highlighting our unique contributions.

We are deeply grateful for your constructive feedback and remain open to addressing any further questions or concerns!

Sincerely,

Authors

---

### Meta-Review · Area_Chair_qziG · 2024-12-22

**Metareview:**

The paper introduces DyAug, a framework that extends graph data augmentation (GDA) to dynamic graphs, addressing the critical challenge of maintaining temporal consistency across graph snapshots. Traditional static GDA methods have limitations in dynamic graph settings, particularly their inability to account for evolving structures over time, which often leads to inconsistencies in temporal representations.

Most reviewers agree on the importance of the problem, the technical novelty of the method (including temporal consistency and dynamic graph augmentations), empirical performance improvement and the presentation. There are some concerns about  scalability, additional baseline comparsions, adversarial settings, ablations, sensitivity analysis, which have been addressed in the rebuttal period.

I would recommend a accept, as supported by most reviewers. However, a more detailed explanation of temporal consistency and the advantages of plug-in-play functionality would improve the paper. The authors are strongly recommended to make revisions according to the reviewers' suggestions.

**Additional Comments On Reviewer Discussion:**

The paper addresses several weaknesses raised by reviewers during the rebuttal period. To improve the scalability concerns, the authors included results demonstrating DyAug's performance on the million-node DGraphFin dataset. They also expanded the evaluation to include additional DyGNN models, reporting results for Roland (KDD 2022) and SLATE (NeurIPS 2024). To address the concern of defending against advanced adversarial attacks, the authors evaluated DyAug’s robustness against the TD-PGD attack, showcasing its ability to withstand more complex adversarial strategies. In response to requests for more comprehensive analysis, the authors conducted additional ablation and sensitivity studies, including experiments under diverse scenarios and a sensitivity analysis for the parameter. Although the evaluation of continuous-time dynamic graphs (CTDGs) was not directly addressed, the authors analyzed the distinctions between DyGNN backbones and GDA methods for CTDGs and DTDGs, highlighting the mutual incompatibility. Lastly, the authors performed a detailed comparison with dynamic graph OOD methods, emphasizing DyAug’s unique contributions in this domain. These revisions enhance the paper’s overall clarity, robustness, and relevance to current challenges in dynamic graph learning.

---

### Decision · Program_Chairs · 2025-01-22

Accept (Poster)